# The seasonal cycle of ice-nucleating particles linked to the abundance of biogenic aerosol in boreal forests

Julia Schneider[1], Kristina Höhler[1], Paavo Heikkilä[2], Jorma Keskinen[2], Barbara Bertozzi[1], Pia Bogert[1], Tobias Schorr[1], Nsikanabasi Silas Umo[1], Franziska Vogel[1], Zoé Brasseur[3], Yusheng Wu[3], Simo Hakala[3], Jonathan Duplissy[3,7], Dmitri Moisseev[3], Markku Kulmala[3], Michael P. Adams[4], Benjamin J. Murray[4], Kimmo Korhonen[5], Liqing Hao[5], Erik S. Thomson[6], Dimitri Castarède[6], Thomas Leisner[1], Tuukka Petäjä[3], and Ottmar Möhler[1]

[1]Institute of Meteorology and Climate Research, Karlsruhe Institute of Technology, Karlsruhe, Germany
[2]Aerosol Physics Laboratory, Physics Unit, Faculty of Engineering and Natural Sciences, Tampere University, Tampere, Finland
[3]Institute for Atmospheric and Earth System Research/Physics, Faculty of Science, University of Helsinki, Helsinki, Finland
[4]Institute for Climate and Atmospheric Science, School of Earth and Environment, University of Leeds, Leeds, UK
[5]Department of Applied Physics, University of Eastern Finland, Kuopio, Finland
[6]Department of Chemistry and Molecular Biology, Atmospheric Science, University of Gothenburg, Gothenburg, Sweden
[7]Helsinki Institute of Physics, University of Helsinki, Helsinki, Finland

*Correspondence to*: Ottmar Möhler (ottmar.moehler@kit.edu)

**Abstract.** Ice-nucleating particles (INPs) trigger the formation of cloud ice crystals in the atmosphere. Therefore, they strongly influence cloud microphysical and optical properties, as well as precipitation and the life cycle of clouds. Improving weather forecasting and climate projection requires an appropriate formulation of atmospheric INP concentrations. This remains challenging, as the global INP distribution and variability depend on a variety of aerosol types and sources, and neither their short-term variability nor their long-term seasonal cycles are well covered by continuous measurements. Here, we provide the first year-long set of observations with a pronounced INP seasonal cycle in a boreal forest environment. Besides the observed seasonal cycle in INP concentrations with a minimum in wintertime and maxima in early and late summer, we also provide indications for a seasonal variation in the prevalent INP type. We show that the seasonal dependency of INP concentrations and prevalent INP types is most likely driven by the abundance of biogenic aerosol. As current parameterizations do not reproduce this variability, we suggest a new mechanistic description for boreal forest environments, which considers the seasonal variation of INP concentrations. For this, we use the ambient air temperature measured close to the ground at 4.2 m height as a proxy for the season, which appears to affect the source strength of biogenic emissions and thus the INP abundance over the boreal forest. Furthermore, we provide new INP parameterizations based on the Ice Nucleation Active Surface Site (INAS) approach, which specifically describes the ice nucleation activity of boreal aerosols particles prevalent in different seasons. Our results characterize the boreal forest as an important but variable INP source and provide new perspectives to describe these new findings in atmospheric models.

## 1 Introduction

Cloud processes are of particular importance for the evolution of weather and climate, as they regulate the global distribution of precipitation and influence Earth's radiative budget (Hoose and Möhler, 2012; Murray et al., 2012). Ice-nucleating particles (INPs) trigger the formation of ice crystals in clouds (Pruppacher and Klett, 2010), and therefore influence cloud microphysical and optical properties, as well as the lifetimes of mixed-phase and ice clouds (Hoose and Möhler, 2012). However, cloud processes remain highly uncertain in weather forecasting and climate projections (Boucher et al., 2013), also due to a lack of understanding of the critical parameters that predict atmospheric INP concentrations. The proportion of aerosol particles, which can act as INPs generally increases with decreasing temperature, as the free energy barrier to nucleation is reduced. Early parameterizations therefore linked INP or primary ice formation in clouds solely to temperature without any link to aerosol

properties (Cooper, 1986; Fletcher, 1962; Meyers et al., 1992). More recent studies suggest a dependence of the INP number concentration on aerosol concentrations in specific size ranges (DeMott et al., 2010; Tobo et al., 2013), air mass origin (McCluskey et al., 2018) and rain events (Huffman et al., 2013; Iwata et al., 2019; Prenni et al., 2013, Stopelli et al., 2015, Stopelli et al., 2017). Others suggest aerosol type-specific descriptions (Harrison et al., 2019; Ullrich et al., 2017; Wilson et al., 2015), e.g. by linking the ice nucleation ability of the aerosol type to the aerosol surface area (Harrison et al., 2019; Ullrich et al., 2017). Due to the abundance of diverse atmospheric INP types distributed over the globe, it is not possible to find a direct dependence of INPs on a single parameter, which could be used to describe and predict primary ice formation processes.

Long-range transport of aerosol particles as well as local sources and sinks influence INP populations and are potentially in flux due to both anthropogenic and natural influences like seasonal cycles. To examine the impact of seasonal changes on the INP population, continuous long-term measurements are necessary, but currently lacking. Only a few studies report atmospheric INP data collected in different seasons and resolve seasonal trends. Hartmann et al. (2019) report INP concentrations from the past 500 years derived from ice core samples collected at two Arctic sites. They do suggest indications that biological INPs contribute to Arctic INP populations throughout the past centuries and assume that it is likely that the strength of local biological particle sources is enhanced during a particular time of the year influencing the INP variability. However, due to the time resolution and dating uncertainty a seasonal relation could not be explicitly shown. Tobo et al. (2019) also report INP concentrations measured at an Arctic station in Svalbard in July 2016 and March 2017, which show seasonal changes with enhanced values in summertime. Tobo et al. (2019) link these enhanced concentrations to the emission of high-latitude dust from glacial outwash plains. Šantl-Temkiv et al. (2019) report INP measurements from the Arctic in spring 2015 and spring and summer 2016 and also show higher INP concentrations in summer than in spring, which they also associate with biological aerosol and biogenic compounds. In another study of pan-Arctic INP, Wex et al. (2019) report INP concentrations from four Arctic stations for different time periods and time resolutions measured between 2012 and 2016. At all locations, the highest observed INP concentrations are recorded in the summer months from June to September. The nature of the INPs was not explicitly determined, but high INP concentrations observed at high activation temperatures indicate a contribution from biogenic material. Wex et al. (2019) suggest potential INP source regions mainly on open land and open water. Stopelli et al. (2015) presents INP concentrations measured from snow samples collected at the Jungfraujoch on a few days per month in the period from December 2012 to September 2013. Here, INP concentrations are again higher in the summer months. Based on this dataset, a model to predict INP concentrations at the Jungfraujoch was established by Stopelli et al. (2016) and validated using several precipitation samples collected between May and October 2014. The dataset from 2014 shows a completely different seasonal pattern than the 2012/2013 dataset with the lowest values during summer and maxima in May and October. The authors suggest these maxima are related to a Saharan dust event and a cold front passage. Schrod et al. (2020) describe a global network of four INP sampling stations, at a range of northern hemisphere latitudes, where atmospheric aerosol samples were collected on substrates for two years from September 2014. The substrates were analysed for deposition and condensation mode INPs using the FRIDGE isothermal static diffusion chamber (Schrod et al., 2020). The Schrod et al. (2020) results do not yield a clear seasonality but instead show that short-term variability overwhelms long-term trends. However, that observation may not represent the full picture of INP in those locations. The short sampling times (low volumes, 1 hour per day) and/or colder activation temperatures may serve to maximize sampling variability and mask any potential biological signal (Schrod et al. 2020). To date these studies present the first observations and indications of the seasonal variability of INP concentrations, addressing the need for more long-term INP observations. None of these studies presents a comprehensive analysis of continuously recorded INP data for a full seasonal cycle at one location without interruptions. Moreover, the focus of most of the previous studies except for the Schrod et al. (2020) study was especially at Arctic INPs.

In our study, we present a long-term record of INP measurements for more than a full seasonal cycle at a remote location in the Finnish boreal forest. The boreal forest ecosystem is one interesting environment for such long-term observations, as INP

measurements in these areas are currently lacking. Boreal environments are characterized by meteorological conditions, vegetation and radiation budgets with strong seasonal trends and a clear annual cycle. Boreal forests cover 15 million square kilometres, representing one-third of all forested land (Tunved et al., 2006). They are generally far from anthropogenic and dust sources and are characterized by high biogenic aerosol concentrations (Kulmala et al., 2013; Spracklen et al., 2008; Tunved et al., 2006). The vegetation in boreal forests emits primary biological aerosol particles (PBAPs) and biogenic volatile organic compounds (BVOCs), which are prone to form secondary organic aerosol (SOA) (Spracklen et al., 2008), and collectively constitute 'biogenic aerosol'. PBAPs are directly derived from biological organisms, for example spores, pollen, fungi and leaf litter, and are distinct from SOA particles that form via new particle formation (NPF) and grow in size by multicomponent condensation (Ehn et al., 2014; Kulmala et al., 2013). BVOCs are integral as precursors for the NPF events, which are frequently observed in boreal forests (Kulmala et al., 1998, 2001). The frequency of NPF events shows a seasonal variability with a bimodal distribution of peak frequencies in spring and in autumn (Dall'Osto et al., 2018; Kulmala et al., 2001; Nieminen et al., 2014). A similar seasonal trend is observed in PBAP concentrations (Manninen et al., 2014; Schumacher et al., 2013).

Several biogenic aerosol types have been shown to have atmospherically relevant ice-nucleating abilities (Augustin et al., 2013; Creamean et al., 2013; Hader et al., 2014; Möhler et al., 2007; Morris et al., 2004; O'Sullivan et al., 2015, 2018; Pratt et al., 2009; Schnell and Vali, 1973) especially at temperatures above -15°C (Christner et al., 2008; Murray et al., 2012). Although the contribution of biogenic INPs to the total global INP abundance is thought to be rather low (Hoose et al., 2010), biogenic aerosol may contribute substantially at regional scales where biological aerosol sources are important. For example, Tobo et al. (2013), Prenni et al. (2009) and O'Sullivan et al. (2018) have observed biogenic aerosol in the INP populations of the forested environments in Colorado, in the Amazon basin and in rural areas in Northern Europe. Furthermore, Pratt et al. (2009) and Creamean et al. (2013) showed that biological particles were frequently present in ice crystal and precipitation residues measured over the western United States and suggested that these particles play a key role in cloud ice formation. In a study about Swedish and Czech birch pollen, Augustin et al. (2013) reported the ice-nucleation activity of sampled macromolecules and formulated new parameterizations for the heterogeneous nucleation rates of two different ice-active macromolecules. However, in general, measuring and parameterizing the IN ability of biogenic particles has proven to be difficult for several reasons. For accurate biogenic INP model simulations, it is critical to understand the global distribution of biogenic INP, their source strength and their aerosolization and atmospheric transport mechanisms (O'Sullivan et al., 2018). It remains unresolved how the microphysical and chemical properties of biogenic aerosol may change during transport processes in the atmosphere. In field studies, which attempt to address these deficiencies, it remains difficult to identify biogenic aerosol particles and to separate them from non-biogenic particles (Möhler et al., 2007). Moreover, there are many biogenic species with a range of properties, which complicate comparisons and generalized parameterizations.

To address this difficulties in a first approach, we systematically measured INP concentrations at the Station for Measuring Ecosystem-Atmosphere Relations SMEARII (Hari and Kulmala, 2005), which is located in the Finnish boreal forest (61° 50' 50.685''N, 24° 17' 41.206''E, 181 m a.m.s.l.). As the nearest city (Tampere) is located about 60 km west-southwest from the station (Sogacheva et al., 2008) the prevalent aerosol population is mainly influenced by the forest. The boreal forest around the SMEARII station is dominated by Scots pine trees (Hari and Kulmala, 2005). In summer 2018, the canopy height of pines at SMEARII was determined to be 21.8 m. An extensive set of permanent measurements at the SMEARII contributes to a well-characterized picture of the site including meteorological as well as general aerosol-related information, which are available on the open research data portal AVAA (Junninen et al., 2009). The first comprehensive ice nucleation campaign at SMEARII called HyICE-2018 took place from February 2018 to June 2018. First results from HyICE-2018 are published in Paramonov et al. (2020), who show INP measurements with a continuous flow diffusion chamber (CFDC) during the first part of the HyICE-2018 campaign from 19 February 2018 to 2 April 2018. Here, we present the results of filter-based INP measurements, which provide a continuous record from 11 March 2018 to 13 May 2018 with a consistent time resolution of

24 hours. After these two intensive sampling months during the HyICE-2018 campaign, the INP measurements were continued until 31 May 2019 with a time resolution of mostly 48 hours or 72 hours, and only in a few cases with sample time intervals of up to 144 hours. By this, we obtained a continuous long-term record of INP temperature spectra from 11 March 2018 to 31 May 2019. The main objective of this study is to investigate and describe the variability and seasonal trends in INP concentrations and INP temperature spectra in a boreal forest environment. The absence of anthropogenic and/or dust aerosol sources in the boreal region motivates the additional investigation of biogenic ice nucleation activity and reveals the relevance of boreal forest areas as an important INP source. The comprehensive instrumentation provided at the measurement site at the SMEARII station allows comparisons between INP measurements with simultaneous measurements of many meteorological variables. These measurements are complemented by measurements characterizing the sampled aerosol number concentrations, size distributions and chemical compositions in order to elucidate the potential origin and nature of the INPs. Heat treatments of the suspensions prior to INP analysis also help identifying the nature of INPs. We aim to improve the parameterizations describing atmospheric INP concentrations in the boreal forest by considering seasonal dependences in the formulations. Finally, this study provides motivation for further continuous long-term studies of INP in different environments across the globe.

## 2 Methods

### 2.1 Aerosol filter sampling

Ambient aerosol particles were collected on 47 mm Whatman nuclepore track-etched polycarbonate membrane filters with a pore size of 0.2 µm. The filter sampling line and the filter holder are made of stainless steel and were installed in a cottage in the forest, with a rooftop PM10 inlet connected to the other sampling components installed indoors. The inlet height is approximately 4.6 m above ground and therefore approximately 17.2 m below the forest canopy. A vacuum pump in combination with a critical orifice ensured a constant sampling flow rate of about 11 std l min$^{-1}$. Because the PM10 inlet provides a precise 10 µm cut-off size for a flow rate of about 16 std l min$^{-1}$, it is possible that larger particles have also been collected. However, the deviation of total aerosol number and surface concentration from the PM10 concentrations is < 1%. We therefore refer to PM10 number concentration and PM10 surface concentration, when comparing the INP concentrations and calculating INAS (Ice Nucleation Surface Site) densities. Filters were pre-cleaned with 10% $H_2O_2$ and rinsed with deionized water that was passed through a 0.1 µm Whatman syringe filter, before being dried for use in sampling. After sampling, the filters were stored in sterile petri dishes, wrapped in aluminium foil and frozen until the sample were analysed for their INP content.

### 2.2 INSEKT

The INP content of the collected aerosol samples was quantified using the INSEKT (Ice Nucleation Spectrometer of the Karlsruhe Institute of Technology) method described in Schiebel (2017). The INSEKT is based on an ice spectrometer developed at the Colorado State University, which is described in Hill et al. (2016). INSEKT measures INP concentration as a function of activation temperature in the immersion freezing mode between about 247 K and 268 K. For the INP analysis, the collected aerosol particles are washed from the filter membranes by immersion in 8 ml of nanopure water, which was passed through a 0.1 µm Whatman syringe filter. The sample solution was spun on a rotator for approximately 20 min, and subsequently the aerosol suspension is diluted with 15- and 225-, or 10- and 100-fold volumes of filtered nanopure water. Small volumes of 50 µl are pipetted into two 96-well PCR plates. The wells are partitioned into different groups, including a group for the undiluted suspensions, the two diluted samples, and for the filtered nanopure water that serves to determine background freezing levels. Filter handling and suspension preparation always occurs in a clean flow cabinet using tweezers, which have been pre-cleaned in the same manner as the filter membranes. The filled PCR plates are then placed into the

INSEKT instrument, which consists of two aluminium blocks, each with openings for holding a 96 well PCR plate. The aluminium blocks are connected to a chiller (LAUDA Proline RP 890), which pumps ethanol cooling liquid through the aluminium blocks at a constant cooling rate of 0.25 K min$^{-1}$. Eight evenly distributed temperature sensors measure the temperature distribution inside the blocks with a 2 Hz resolution. The aluminium blocks are placed in a PVC box insulated with 2 cm of Armaflex insulation material. The upper part of the PVC box is equipped with an antireflection and depolarized glass pane covering the PCR plates and preventing contamination from the ambient air. In order to avoid condensation on the glass the interior of the PVC box and the upper side of the glass pane are continuously flushed with particle free synthetic air at a constant flow rate of about 80 l h$^{-1}$. A camera with a 60 cm focal distance detects brightness changes in the small suspension volumes that are related to freezing during the cooling process. LabView software is used to control and monitor the cooling rate, temperature and brightness changes. Using this setup the frozen fraction of the small aerosol suspension volumes are determined as a function of temperature. From the fraction frozen, the INP concentration per standard litre of sampled air is calculated, binned on a 0.5 K grid and corrected by the background from the filtered nanopure water, using the procedure described in Vali (1971). Error is estimated by determination of 95% confidence intervals using the Wilson score interval (Wilson, 1927) in the form described by Agresti and Coull (1998). Data points with a ratio of upper to lower confidence interval higher than 8 are considered insignificant and neglected. A systematic error due to the preparation process and flow measurements is added. Applying a simple linear error propagation on the formulas given in Vali (1971) and inserting the error-containing parameters like the pipetted suspension volumes and the flow rate systematic errors of 4% for the undiluted suspension, 5% for the first dilution step, 8% for the third dilution step and 11% for the fourth step, are calculated. The systematic error increases with each dilution step because the additional pipetting step adds uncertainty. In addition, INP concentrations derived from handling blank filters, which were collected without flowing air through the membranes, are subtracted. Heat treatment tests of the collected aerosol samples provide additional information about the heat sensitivity of the containing INPs and have been applied in various previous INP studies (Hill et al., 2016; O'Sullivan et al., 2018; Wilson et al., 2015). For these tests, a test tube filled with 2 ml of the aerosol suspension is kept in boiling water for approximately 20 min. Afterwards, the treated sample is analysed with the INSEKT in the same way as previously described. INAS densities were calculated as described in Eq. (2) in Ullrich et al. (2017), where ice number concentrations are normalized by the aerosol surface area concentration. Assuming that every INP triggers the formation of one ice crystal, the ice number concentrations are equal to the INP concentrations, which are determined by the INSEKT measurements. The aerosol surface area concentrations are derived from continuous size distribution measurements of the PM10 atmospheric aerosol at SMEARII. Details on the size distribution measurements are given in the following section.

## 2.3 Additional Instrumentation at SMEARII

For characterizing the sampled aerosol particles further, atmospheric aerosol size distributions were continuously measured with a Differential Mobility Particle Sizer (DMPS). The covered size range was 3 nm - 1000 nm in electrical equivalent diameter with 10 min time resolution (Aalto et al., 2001) with a closed loop flow arrangement (Jokinen and Mäkelä, 1997). The instrument was operated following guidelines from Aerosols, Trace Gases, and Clouds Research Infrastructure (ACTRIS) (Wiedensohler et al., 2012). The aerosol sample was taken from 8 m height inside the canopy through a total suspended particle (TSP) inlet. The super-micron aerosol size distribution was determined with a TSI Aerodynamic Particle Sizer (APS) model 3321 for the size range 0.5 - 10 µm in aerodynamic diameter. The sample was drawn through a vertical sampling line to avoid particle losses. The inlet is at a height of 6 m above the ground and consists of a total suspended particle inlet (Digitel Inc.). The inlet was heated to 40°C to prevent condensation and to ensure that fog droplets are evaporated and the RH remains below 40 %.

The intense measurement period during the HyICE-2018 campaign also provides additional aerosol instrumentation, like a long time-of-flight aerosol mass spectrometer (L-ToF-AMS) to measure the aerosol chemical composition and a wideband integrated bioaerosol sensor (WIBS-NEO) to derive information about biogenic fluorescent aerosol particles. The WIBS-NEO (Droplet Measurement Technologies, Longmont, CO, USA) is a bioaerosol sensor that provides information on the fluorescence properties, size and asphericity ratio of individual aerosol particles. It operates with an inlet flow of 0.3 l min$^{-1}$ and detects particles with diameters between 500 nm and 30 µm. From 11 March 2018 to 2 April 2018, the WIBS was located about 50 m from the aerosol filter sampling line used for the INP analysis. There, it was attached to a total aerosol inlet, which is characterized in Vogel (2018). On 3 April 2018, the WIBS was moved and installed directly next to the filter sampling line and attached to a PM10 inlet, which is described in Schmale et al. (2017). For the WIBS data analysis, particles from 0.5 µm to 10 µm were considered. To analyse the fluorescence of the particles, the WIBS sensor utilizes two xenon flashlamps as excitation light sources (optically filtered at wavelengths of 280 nm and 370 nm) and two emission detection channels (wavelength bands 310 – 400 nm and 420 – 650 nm). Optical size information is acquired utilizing elastic scattering from a continuous wave laser with a wavelength of 635 nm and a photomultiplier tube located orthogonally with respect to the laser. The excitation pulses are fired into the sample volume at different times and both detection channels record the emission(s) from both excitations, leading to three distinguishable excitation-emission combinations (the 370 nm light saturates the 310 – 400 nm detection channel and therefore does not provide any information). Thus, the fluorescence can be divided into 7 unique fluorescence groups based on the excitation-emission wavelength pairs and their combinations after Perring et al. (2015) and Savage et al. (2017): A (only FL1: excitation 280 nm, emission 310 – 400 nm), B (only FL2: excitation 280 nm, emission 420 – 650 nm), C (only FL3: excitation 370 nm, emission 420 – 650 nm), AB (FL1 + FL2), BC (FL2 + FL3), AC (FL1 + FL3) and ABC (FL1 + FL2 + FL3). The WIBS performs an empty-chamber background signal check every 8 hours, during which the excitation pulses are fired into the optical chamber without any present particles. The background check collects a multitude of emission intensities that form a baseline for particle fluorescence. In this study, a particle is considered fluorescent, if the associated emission peak intensity is larger than $FT + 9\sigma$. $FT$ is the mean value of the forced trigger intensities and $\sigma$ is their standard deviation. A more commonly used method would be to compare the emission peak intensity to $FT + 3\sigma$. However, some non-biological particle types such as wood smoke, African dust and black carbon are weakly fluorescent and therefore might satisfy the lower threshold value, leading to an overestimation of biological particle concentration. Furthermore, the stricter threshold only marginally affects the detection efficiency of biological particles, because they tend to have stronger fluorescence (Savage et al., 2017). More detailed descriptions on the WIBS are also available in Savage et al. (2017) and Perring et al. (2015). Daily size distributions measured by the DMPS and APS combination are compared with the size distributions measured by the WIBS from 11 March 2018 to 13 May 2018 and are shown in Figure A3. In summer, WIBS tends to measure slightly more particles with diameters larger than about 3 µm compared to the APS. However, the size distributions agree well for the other time periods and the smaller size ranges.

The size-resolved chemical composition of ambient aerosol was measured with the L-ToF-AMS. Its application in the same campaign has been described in Paramonov et al. (2020). It builds on the functionality and characteristics of the high-resolution ToF-AMS (DeCarlo et al., 2006). However, due to the longer time-of-flight chamber, the L-ToF-AMS, has a better resolution (8000 M/ΔM) than the standard ToF-AMS (2000 M/ΔM in V-mode). Detailed descriptions of the instrument, measurements and data processing are available in other publications (Canagaratna et al., 2007; DeCarlo et al., 2006). In general, the L-ToF-AMS measures the size-resolved, non-refractory composition of submicron aerosols, including organic, sulfate, nitrate, ammonium and chloride. The aerodynamic lens has a 100% transmission range of 75-650 nm (in vacuum aerodynamic diameter; Liu et al. (2007)) and focuses particles into a narrow beam that impacts the surface of a porous tungsten vaporizer heated to 600°C, followed by ionization by a 70eV electron source. Ions are detected by a long time-of-flight mass analyzer (Tofwerk AG). The sample flow of 0.09 l min$^{-1}$ is extracted from an extra suction flow (3 l min$^{-1}$) that is used to avoid aerosol losses in the inlet line. A PM2.5 cyclone mounted at the inlet removes large particles to avoid clogging the critical orifice

(100μm), and before entering the L-ToF-AMS, the samples are dried by a Nafion dryer to keep the RH below 30%. The L-ToF-AMS data were analyzed using standard ToF-AMS data analysis toolkits (Squirrel V1.61B and PIKA1.21B) using Igor Pro software (V6.37, WaveMetrics Inc.). To calculate mass concentrations an ionization efficiency (IE) was determined using 300 nm, size-selected, dry ammonium nitrate particles, and a relative ionization efficiency (RIE) for ammonium of 3.7 was determined. The default relative ionization efficiency (RIE) values of 1.1, 1.2, 1.3 and 1.4 for nitrate, sulfate, chloride and organics, respectively, were applied. A composition-dependent collection efficiency (CE) was applied based on the principles proposed by Middlebrook et al. (2012).

Various meteorological parameters are continuously monitored at SMEARII. For this study, we used five basic variables including ambient air temperature, relative humidity, wind speed, snow depth and precipitation. The ambient air temperature was measured 4.2 m above ground with a Pt100 sensor inside a ventilated custom-made radiation shield. This 4.2 m temperature measurement is the closest to ground-level at SMEARII and thus we utilize this as the ground-level ambient air temperature in the following. The relative humidity was measured in 35 m height by a Rotronic MP102H RH sensor. For wind speed measurements, we used a Thies 2D Ultrasonic anemometer at 34 m above the ground by. The snow depth was measured by a Jenoptik SHM30 snow depth sensor, which is based on an opto-electronic laser distance sensor, in open field about 500 m southeast of the aerosol collection area of SMEARII. The precipitation, the liquid water equivalent, was measured by a Vaisala FD12P Weather Sensor in 18 m height.

## 3 Results and Discussion

### 3.1 INP temperature spectra and time series

All INP temperature spectra measured from 11 March 2018 to 31 May 2019 are shown in Figure 1 in a monthly representation. The INP concentrations range from about $10^{-4}$ std l$^{-1}$ to $10^{-2}$ std l$^{-1}$ at the highest and from about $10^0$ std l$^{-1}$ to $10^2$ std l$^{-1}$ at the lowest temperatures. These concentration values fall within the range of INP concentrations measured during previous globally distributed field studies, which are summarized in Kanji et al., (2017). This indicates that primary ice formation in boreal forest areas is comparable to other regions on Earth, despite the lack of anthropogenic and dust sources. In our study, we observe both INP concentrations and spectral shape to be highly variable from day to day and to show clear seasonal trends. INP temperature spectra in March, December, January and February constitute the lowest of the entirety of INP temperature spectra, whereas the INP temperature spectra with highest INP concentrations are recorded in May and September. Figure 2a depicts the full time series of INP concentrations measured as a function of activation temperature with a time resolution between 24 h and 144 h. The colour contours represent the seasonal cycle of INP concentrations, which are lowest in wintertime from December to March and highest in the summer months, especially during May and September, as it was already observed in Fig. 1. An additional peak is found in the beginning of July. The INP concentrations in the middle temperature range around 257 K show the most distinct seasonal cycle. The variability of INP concentrations at the lower and upper end of the temperature range is less pronounced. Figure 2b shows the time series of INAS densities, which are calculated by normalizing the INP concentration measured by INSEKT with the aerosol surface concentration of atmospheric PM10 aerosol particles derived from DMPS and APS. The INAS densities show the similar seasonal trend and annual variability as the INP concentrations. For comparison, the time series of PM10 aerosol number concentrations and PM10 aerosol surface concentrations are shown in the Appendix Fig. A1.

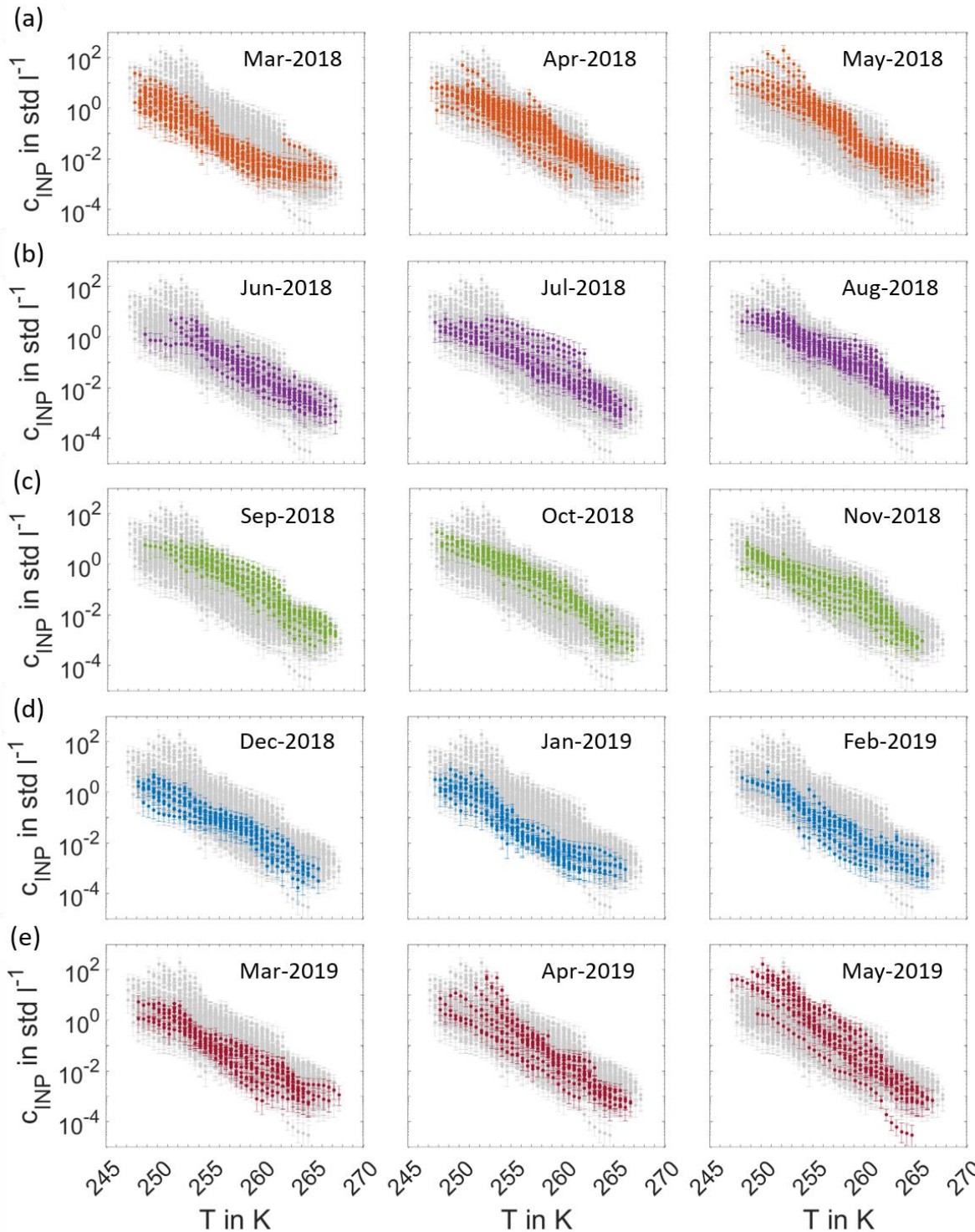

290

**Figure 1: Monthly overview of INP temperature spectra.** Each panel shows the entirety of INP temperature spectra measured from 11 March 2018 to 31 May 2019 (grey) with the spectra of the specific month highlighted in colour.

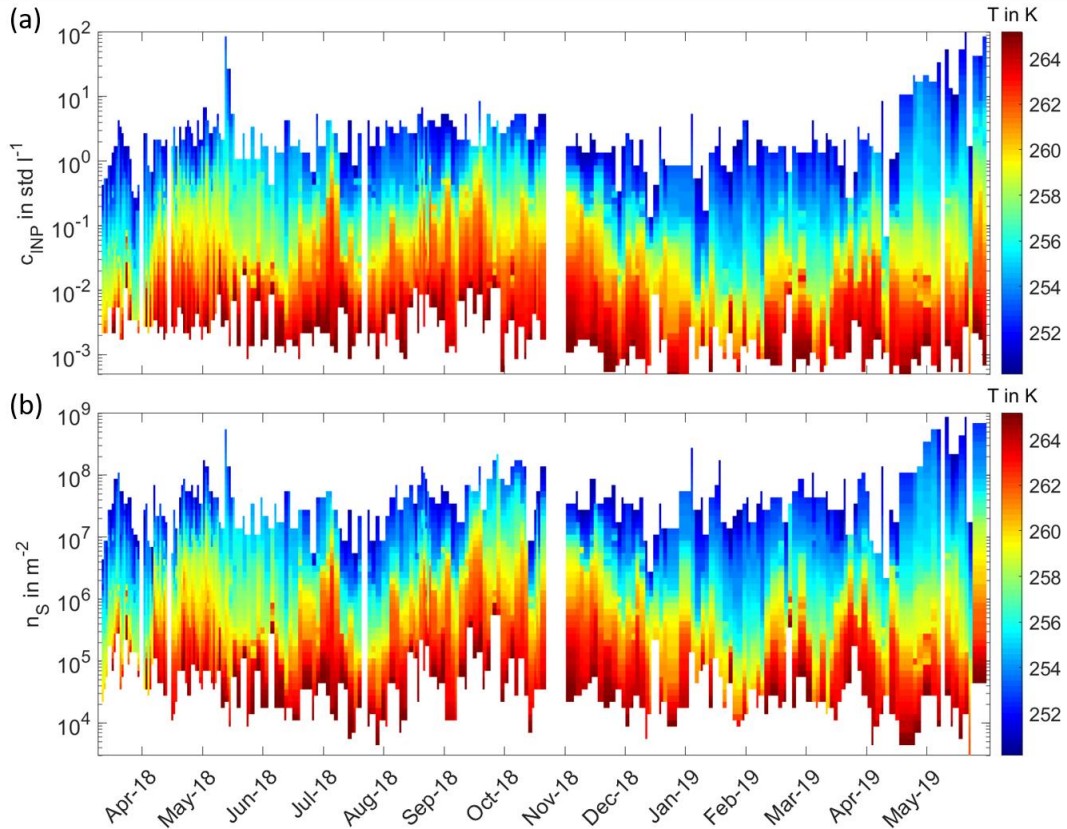

**Figure 2: Long term record of INPs from 11 March 2018 to 31 May 2019 in the Finnish boreal forest. Panel (a) shows the time series of INP concentrations $c_{INP}$ with a general temporal resolution between 24h and 144h. The colour code indicates the corresponding activation temperatures in K. In panel (b) the time series auf INAS densities $n_S$ is displayed in the same manner as for the INP concentrations in panel (a).**

### 3.2 Comparison to meteorology and aerosol properties

To investigate factors that influence the abundance of INP and might explain the daily and seasonal variability of INP concentrations, the INP time series are compared with other data sets like meteorological and aerosol data. As the seasonal trends and variability is most pronounced at activation temperatures around 257 K, the INP time series at 257 K is used for this comparison. In Fig. 3, the monthly averaged INP time series at 257 K is compared with the monthly fraction of NPF event days and snow coverage measured alongside the INP measurements at SMEARII, as well as with the averaged concentrations of pollen and other PBAP. We have defined snow coverage as measured snow depth > 1 cm. The analysis of NPF events is based on permanent measurements at SMEARII and was provided by Simo Hakala, University of Helsinki. As there were no simultaneous direct measurements of pollen and other PBAP available for the period of our INP measurements, we compare to pollen and PBAP measured in 2003 and 2004 by Manninen et al. (2014) at SMEARII. Manninen et al. (2014) collected aerosol samples in a Hirst-type volumetric spore trap (Burkard Manufacturing Co. Ltd.; Hirst, 1952), located at SMEARII 3 m above the forest canopy. The trap is driven by a clockwork and collects aerosol particles larger than approximately 3 μm on an adhesive, transparent, plastic tape with a sampling flow rate of approximately 10 l min$^{-1}$. The analysis of the collected particles was performed according to standard methodology adopted by the Finnish pollen information network and following the principles of the European Aeroallergen Network (www.polleninfo.org/) and Rantio-Lehtimäki et al. (1994). The observed INP peak in spring coincides with the peak in pollen concentrations, whereas the peak in September is found to correlate with enhanced concentrations of other PBAP. Maxima in NPF event fractions are recorded in spring and in autumn, which has also been observed in many previous years back to 1996 (Dall'Osto et al., 2018; Kulmala et al., 2004; Nieminen et al., 2014). Snow-free periods are characterized by relatively high INP concentrations, whereas complete snow cover yields low concentrations.

Figure 4 shows the INP time series at 257 K focussing on the intensive measurement period of the HyICE-2018 campaign from March to May 2018. After a period of rather constant INP concentrations in March, we observe a steady increase of INP concentrations in April, which comes along with the snowmelt period at SMEARII. After the snowmelt, INP concentrations are again on a rather constant but higher level. During the HyICE-2018 period, more comprehensive aerosol characterization was done, including measurements with the L-ToF-AMS and the WIBS shown in Fig. 4a and b. Here, we define the number concentration of fluorescent particles as the number concentration of aerosol particles, whose fluorescence emission intensity produces a fluorescent signal in the fluorescence group ABC (see Section 2.3 for details on the categorization in fluorescence groups). The time series of the number concentration of particles with a fluorescence signal in other fluorescence groups is shown in the Appendix in Fig. A2. In this Figure, the strongest seasonal increase in the transition period from winter to summer is observed in the group ABC. Consequently, this fluorescent group correlates best with the measured INP concentrations (see Fig. 4a). The characteristics of each fluorescence group are comprehensively investigated and reported in Savage et al. (2017), who examined the fluorescence emissions of different types of pollen, fungi, bacteria, biofluorophores, dust, HULIS (humic-like substances), PAH (polycyclic aromatic hydrocarbons), soot and brown carbon. Using the $FT + 9\sigma$ threshold for defining a particle as fluorescent, nearly all dust and HULIS types show no fluorescence signal at all. Some of the soot and brown carbon types only show weak signals in A, and B, BC and A, respectively. Nearly all of the bacteria types show fluorescence only in group A. The fluorescence of fungal spores are also mainly detected in group A, but also in AB and ABC. The investigated biofluorophores show mainly fluorescence in the groups BC (Riboflavin, NAD), A (Pyridoxamine), AB (Tryptophan) and ABC (Ergosterol). PAHs show fluorescence mostly in groups ABC and A. Finally, most pollen types show fluorescence in groups ABC and AB. Some pollen types also show a fluorescence signal in groups A and B. Both, the organic aerosol mass concentration measured by the L-ToF-AMS and the number concentration of fluorescent particles measured by WIBS, tend to increase during the transition period from winter to summer (Fig. 4a and b). The number concentration of atmospheric PM10 aerosol measured by APS/DMPS (Fig. 4c) does not show this trend. However, a slight seasonal trend is visible in the PM10 surface concentration (Fig. 4d). The number and surface concentration of PM10 atmospheric aerosol for the whole time period from March 2018 to May 2019 is shown in the Appendix in Fig. A1.

Figure 5a depicts the whole time series of INP concentrations in comparison to the measured ground-level ambient air temperature from March 2018 to May 2019, which clearly shows the INP time series a following the course of the ground-level ambient air temperature. Only for ground-level ambient air temperatures about > 15°C, the deviation of the two time series is higher. However, the general seasonal trend in the ground-level ambient air temperature is the same as we observed in the INP time series. Besides a significant correlation of INP concentrations to the ground-level ambient air temperature, the snow depth also showed a significant correlation. Figure 5b shows a direct comparison of the INP time series with the measured snow depth. Periods of decreasing INP concentrations clearly overlap with snowmelt periods, whereas an increasing snow depth comes along with increasing INP concentrations. Figure 5c shows the comparison of the INP time series with the time series of relative humidity (RH) measured 35 m above ground. Over the entire time period, no clear relationship between RH and INP concentrations is observed. For a shorter time period from June 2018 to September 2018, there seems to be some correlation of the INP concentration with the RH, during which the peaks in INP concentration in June corresponds to a RH peak. In Figure 5d, we compare the time series of INP concentrations with the time series of wind speed measured 34 m above the ground, which is also above the forest canopy. A relationship between measured INP concentrations and wind speed is not observed. In Figure 5e, the time series of INP concentration is compared to the occurrence of precipitation. Although other studies like Prenni et al. (2013), Huffman et al. (2013) and Iwata et al. (2019) report increasing INP concentrations during and after rain events in forested sites, we do not consistently observe this behavior. In this respect, increased INP concentrations are observed only during two of the strongest precipitation events in June and September 2018. However, it should be noted that in the cited studies INP concentrations were measured with higher time resolutions from minutes to hours. Huffman et al. (2013) reported increased INP and biological particle concentrations during rain events and up to one day after rain events.

With our sampling (and therefore averaging) time of 24 hours or more, rain-induced enhancements of INP concentrations may have been missed. Therefore, this type of sampling strategy may not be appropriate to deterministically link INP concentrations with rain events.

In wintertime, complete snow cover seems to suppress biogenic particle emissions, resulting in comparably low INP concentrations. Such a correlation is also supported by the peaks of pollen and PBAP concentrations in snow-free periods in spring and in autumn, and by the increases of the organic aerosol mass concentration and fluorescent particle numbers of group ABC observed in spring. According to the study of Savage et al. (2017), we assume particles of fluorescence group ABC to be mainly pollen, particles containing PAH or Ergosterol, or fungal spores. The non-refractory organic components measured by AMS include the commonly observed primary organic aerosol (POA) and oxygenated organic aerosol (OOA). As the surface concentration of PM10 particles is more sensitive to larger particles, the increase in PM10 surface concentration (see Fig. 4d) indicates that the observed seasonal increase may be due to larger particles, which are expected to be mainly of biogenic origin. No day-by-day relation of NPF events and INP peaks are identified, but this is not unexpected given that particles formed during NPF events are initially smaller than 5 nm in diameter and events are more likely when condensation sinks are low (Dada et al., 2017). However, we consider the enhanced NPF event frequency as an indicator of generally higher biological activity in the forest. Enhanced biological activity means more biogenic INP emissions from the vegetation, which agrees with the seasonal dependencies from the previous findings. Any direct impact of NPF events on the boreal forest INP abundance remains uncertain and requires more investigation.

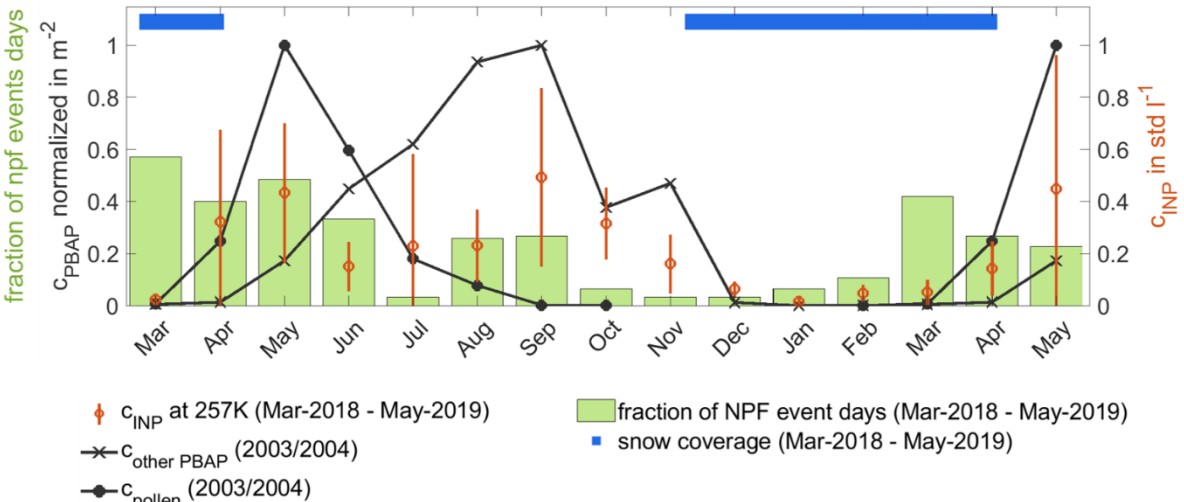

**Figure 3: Factors co-varying with INP concentrations. Monthly averaged INP concentrations at 257 K (red, standard deviation in error bars) are compared to the monthly fraction of NPF event days (green bars) from March 2018 to May 2019. Monthly averaged concentrations of PBAP and pollen (black dots and crosses) measured in 2003 and 2004 by Manninen et al. (2014) are additionally displayed. The blue bar is indicative of snow coverage.**

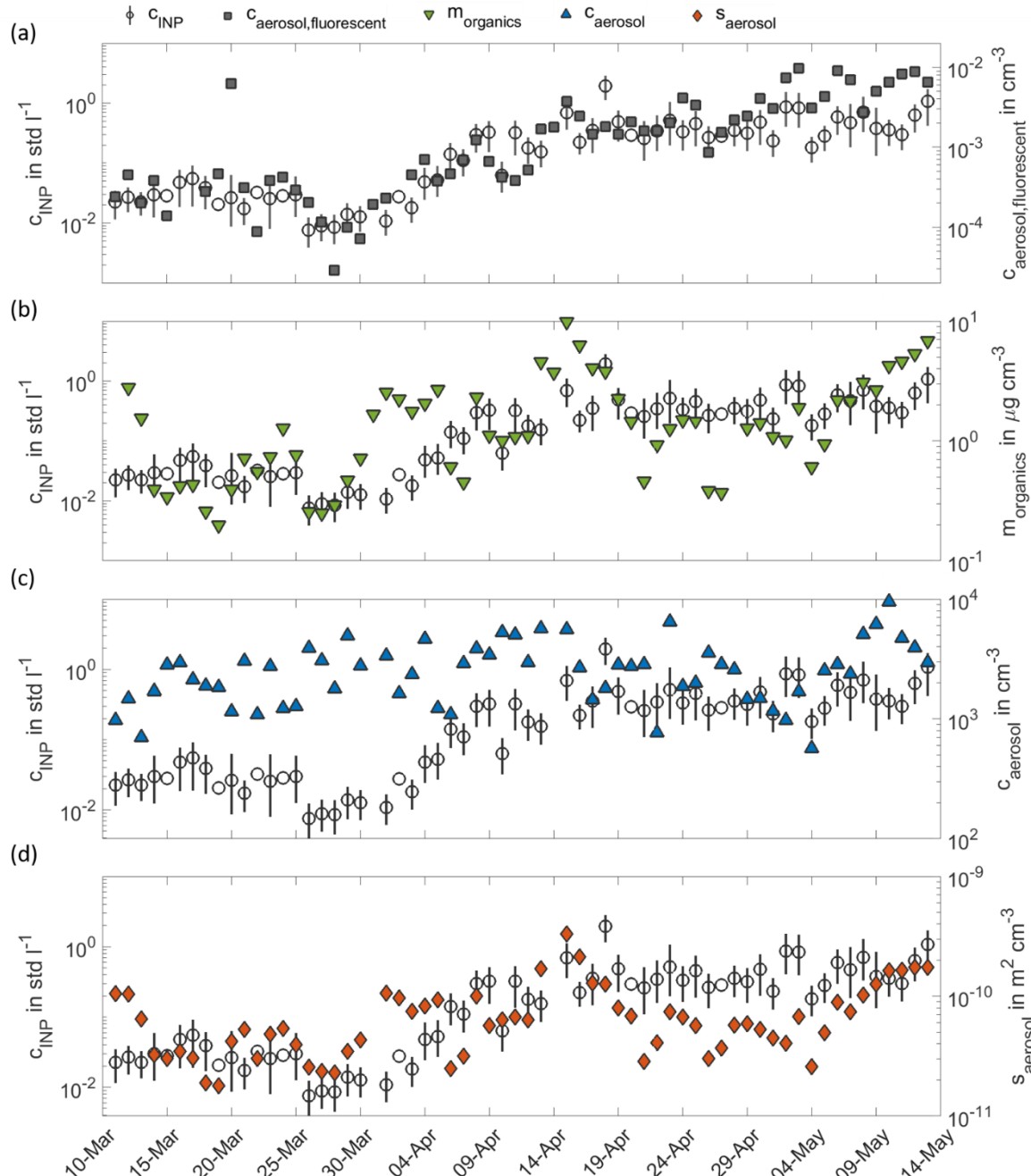

**Figure 4: Factors co-varying with INP concentrations in the HyICE-2018 period. In panel (a), INP concentrations at 257 K (circles) are compared to the concentration of fluorescent aerosol particles with a fluorescence signal in group ABC (grey squares). In panels (b), (c) and (d) the INP concentrations are further compared to the mass concentration of non-refractory organic compounds (green triangles), the number concentration of atmospheric PM10 aerosol (blue triangles) and the surface concentration of atmospheric PM10 aerosol (orange diamonds).**

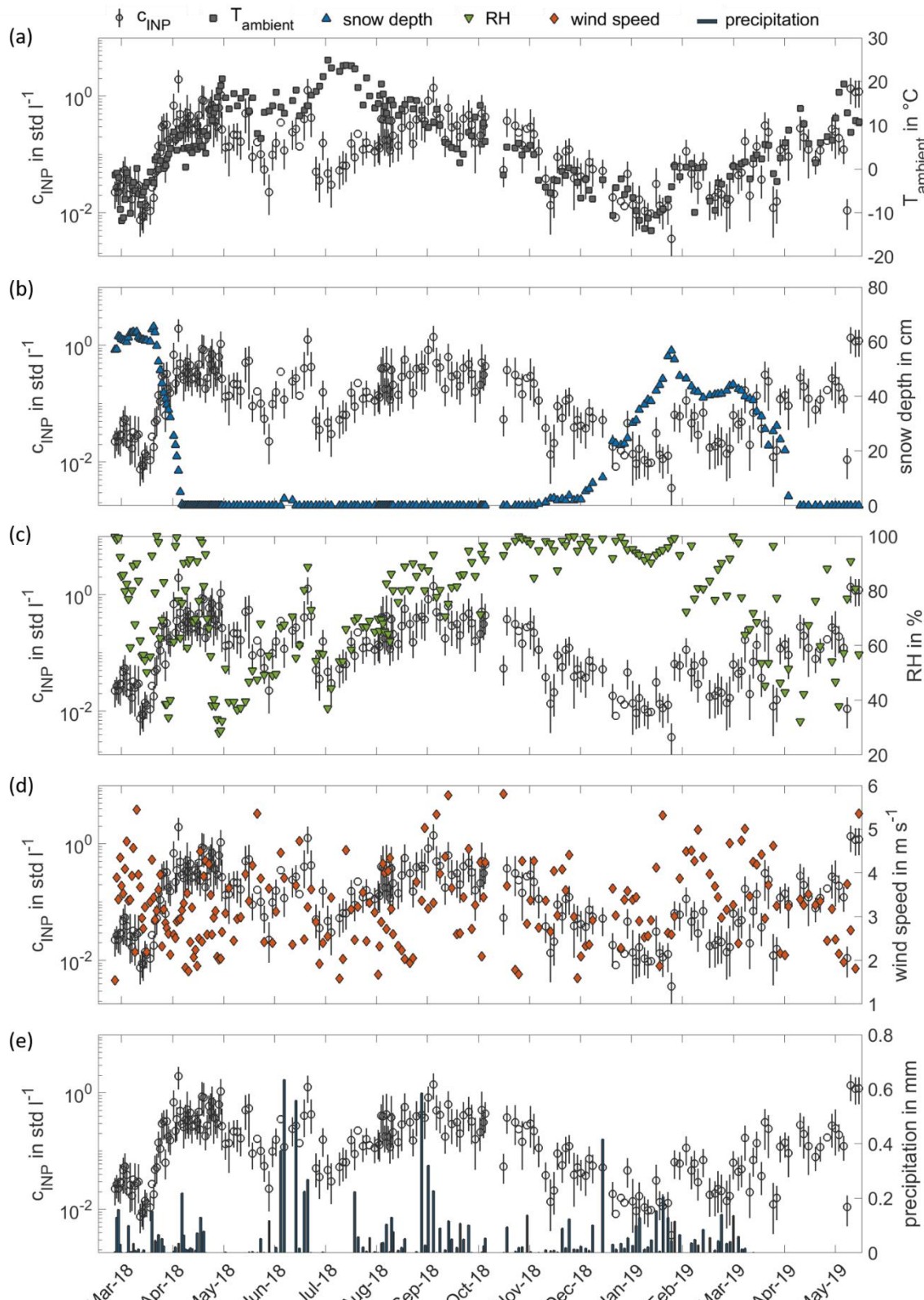

**Figure 5: Time series of INP concentrations compared with the time series of different meteorological parameters from 11 March 2018 to 31 May 2019. In panel (a), the time series of INP concentrations at 257 K (circles) is plotted with the ground-level ambient air temperature measured at 4 m above ground level (grey squares). In panel (b), (c), (d) and (e) the INP concentrations at 257 K are replotted and compared to the measured snow depth (blue triangles), the relative humidity measured in 35 m height (green triangles), the wind speed measured in 34 m height (orange diamonds) and precipitation (black bars). All meteorological parameters are averaged over the INSEKT filter sampling time intervals.**

## 3.3 Heat treatment tests

The exemplary INP spectra in Fig. 6 demonstrate that the INP concentrations show an exponential trend with activation temperature (approximately linear shape of the $\log(c_{INP})$-T-spectra) during wintertime, whereas summertime spectra show enhanced concentrations at around 260 K resulting in curvature in the spectra. After heating the INSEKT samples in boiling water for approximately 20 min, the resulting INP spectra are shifted towards lower concentrations by one to two orders of magnitude throughout the temperature range. However, the characteristic bulge in the summertime spectra is conserved or even more pronounced after the heat treatment. The observed shift of INP spectra after heat treatment reveals the presence of heat-labile INP types, which hints to particles of biogenic origin containing ice active proteinaceous material (Hill et al., 2016; Morris et al., 2004). As a significant number of residual heat-resistant INPs is still remaining after heat treatment, this indicates that not all measured INPs are associated with heat-labile biogenic materials. However, the majority of the INP population seems to be dominated by heat-labile materials, which is shown by the systematic shift in INP-temperature spectra. The characteristic differences in the shapes of the INP spectra further indicate that different aerosol types dominate the INP populations in winter- and summertime. These differences are consistent for the observations in 2018 and in 2019, suggesting a systematic seasonal behaviour.

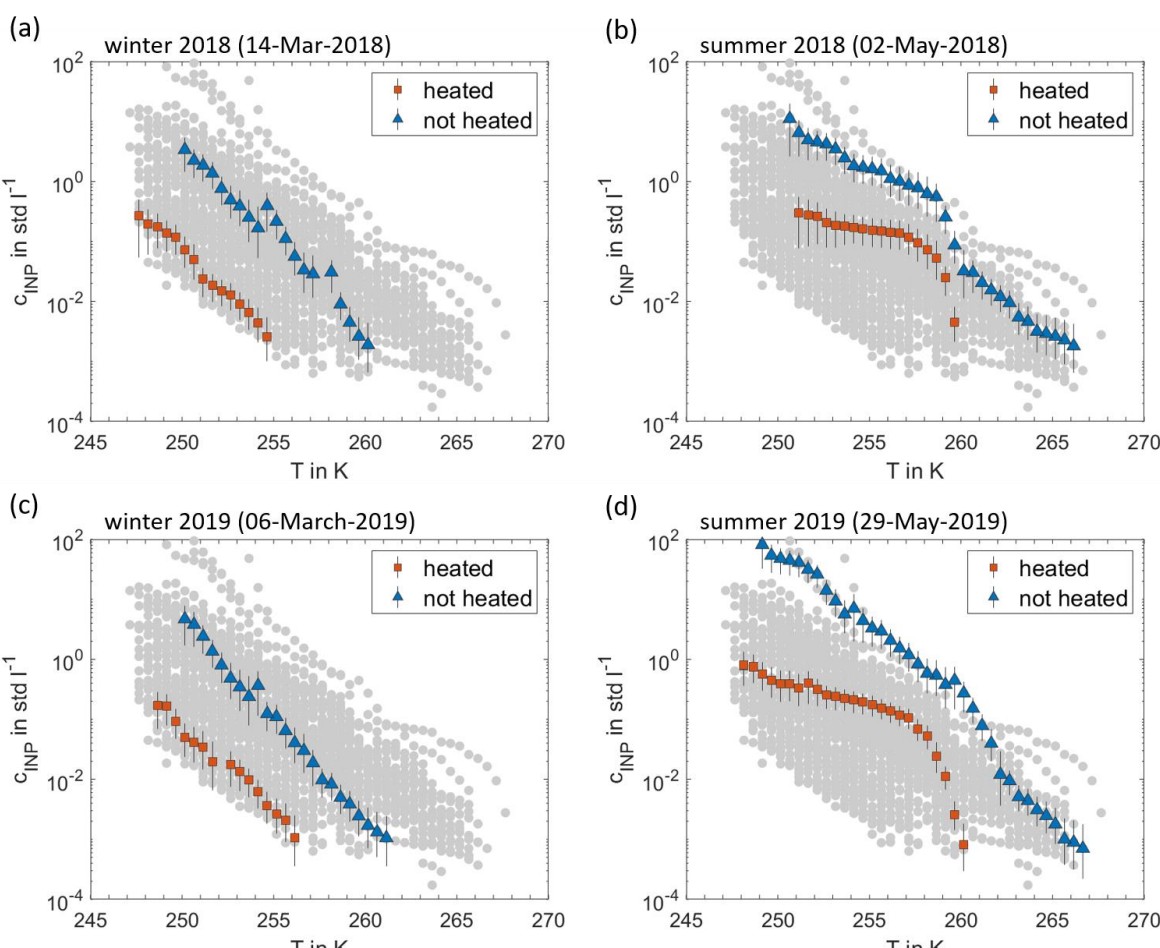

**Figure 6: Effect of heat treatments. INP temperature spectra of the non-heated aerosol samples (blue triangles) are compared to the spectra of the heat-treated samples (red squares). Examples of two days in 2018 typical for winter and summer (panels (a) and (b)) and two days in 2019 also typical for winter and summer (panels (c) and (d)) are shown. In each panel, the grey dots display the entirety of non-heated and heat-treated samples.**

## 3.4 Empirical parameterizations

The observations presented in Fig. 1-6 indicate that the INP populations in boreal environments are dominated by biogenic emissions from the vegetation in the forest. We provide evidence that INP concentrations experience a seasonal cycle, which

we link to seasonal trends in biogenic aerosol. The observational data we have presented poses new challenges for quantitative INP predictions, as it is essential to incorporate seasonal trends to achieve accurate descriptions. In Fig. 7a-d the measured INP concentrations are plotted versus the INP concentrations predicted by current parameterizations (DeMott et al., 2010; Tobo et al., 2013; Ullrich et al., 2017). DeMott et al. (2010) and Tobo et al. (2013) have developed temperature-dependent parameterizations that use the number concentration of aerosol particles with diameters > 0.5 µm (Fig. 7a and b). Tobo et al. (2013) provide an additional temperature-dependent formulation using the FBAP (fluorescent biological aerosol particle) concentration (Fig. 7c). Ullrich et al. (2017) use the measured aerosol surface area concentration to formulate a temperature-dependent parameterization of the INAS density of mineral dust (Fig. 7d). Among the selected parameterizations, Tobo et al. (2013) reproduce most of the data points by predicting 63% (Fig. 7b) and 80% (Fig. 7c) of the measurements to within one order of magnitude. Note for the application of the Tobo et al., (2013) parameterization using FBAP concentrations, only data from the HyICE-2018 time period could be used, as the fluorescence measurements from the WIBS are only available in this period. Therefore, the number of data points in Fig. 7c is lower than in the other panels. The aerosol-specific parameterization of Ullrich et al. (2017) best matches the temperature trend, but overestimates the measured INP data, reproducing only 23% of the data points to within a factor of 10. This is not surprising, as the boreal forest aerosol is not dominated by mineral dust. The predictions of DeMott et al. (2010) and Tobo et al. (2013) overestimate INP concentrations especially in wintertime. These comparisons emphasize the need for a parameterization that accounts for seasonality.

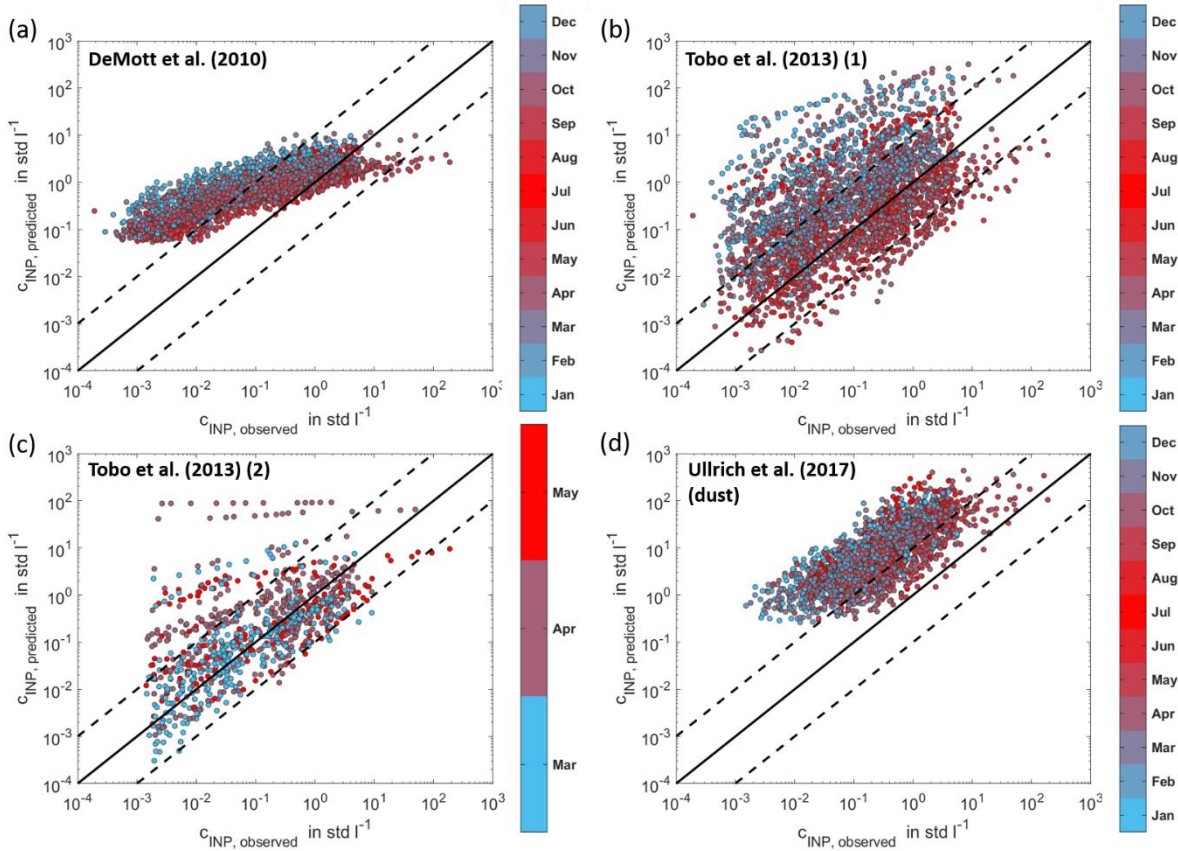

**Figure 7: Comparisons to INP parameterizations. The measured INP concentrations are compared to INP concentrations predicted by parameterizations from DeMott et al. (2010) (a) and Tobo et al. (2013) (b) both using the number concentration of aerosol particles with diameter > 0.5 µm. Further, the measured concentrations are compared to the predicted concentrations by the second parameterization of Tobo et al. ( 2013) using the number concentration of FBAP (c) and to the parameterization by Ullrich et al. (2017) for the temperature-dependent INAS density of mineral dust (d). In Tobo et al. (2013), FBAP concentrations were measured using an excitation wavelength of 355 nm and detecting the fluorescence emission in the range of 420 – 575 nm. The black lines show the 1-1-line (solid) and the 1-10-line and 10-1-line, respectively (dashed).**

Given the need to represent seasonality, we suggest two new formulations for describing the seasonal variability of boreal forest INPs. Our first new non-aerosol specific approach assumes that the atmospheric INP concentration is predominantly determined by the boreal forest as the major INP emitting source with a magnitude that naturally changes with the seasons.

Using ground-level ambient air temperature averaged over the aerosol filter sampling times as a proxy for the seasonal cycle, the observed INP concentration is quite well overlaid (see Fig. 5a). This clear relationship motivates us to use this parameter for formulating the new parameterization. The measured INP temperature spectra between 250 K and 265 K were used to create a least squares fit between the activation temperature T in K and the natural logarithm of INP concentrations $c_{INP}$ in std l$^{-1}$. This describes the activation behaviour of INPs in temperature space. To account for seasonal dependencies in this

formulation, the linear relation between the ambient air temperature $T_{amb}$ in K measured close to the ground at 4.2 m height (called ground-level ambient air temperature) and the natural logarithm of the time series of INP concentrations $c_{INP}$ in std l$^{-1}$ was used to establish a prefactor which shifts the parameterized INP temperature spectra to higher or lower INP concentrations depending on the ground-level ambient air temperature. The resulting parameterization is

$$c_{INP} = 0.1 \cdot exp\,(a1 \cdot T_{amb} + a2)\, \cdot exp\,(b1 \cdot T + b2)\, std\; l^{-1} \tag{1}$$

with a1 = 0.074 ± 0.006 K$^{-1}$, a2 = -18 ± 2, b1 = -0.504 ± 0.005 K$^{-1}$, b2 = 127 ± 1 and with the activation temperature T and ground-level ambient air temperature $T_{amb}$ in K (measured at 4.2 m height). This new parameterization is able to reproduce 97.22% of the data to within a factor of 10. 88.21% and 49.79% are reproduced within a factor of 5 and 2, respectively. In Fig.

8a measured INP concentrations are compared to those predicted by the new parameterization underlining the good agreement with a goodness of fit of R² = 0.82. This new parameterization approach describes the annual variation of the near-surface INP concentration in the boreal forest, which provides a temperature dependent source of these INPs. We did not directly detect or quantify the INP source, but found a strong correlation of the measured INP concentration with the ground-level ambient air temperature. This leads to the assumption that the near-surface INP concentration in the boreal forest may be dominated by

local or regional sources, and that this parameterization may be used in models to formulate the source strength or concentration of INPs in boreal forests near the surface. It should be noted that this is a mechanistic approach, which cannot necessarily be applied to regions other than boreal areas or to higher altitudes, where the INP spectrum may be influenced by other sources. It is further important to note that INPs might undergo changes in their size distribution and chemical composition, when they are transported from their sources to higher altitudes, which could affect their ice nucleation ability.

Our second formulation is aerosol-specific and describes the ice nucleation efficiency of boreal forest aerosol types using the INAS approach (Ullrich et al., 2017; Vali, 1971). As the seasonal cycle of INAS densities (see Fig. 2b) and the heat test results described earlier (see Fig. 6) indicate a seasonal change in INP types, we suggest specific parameterizations for different seasons. For the INAS density parameterization for boreal forest INPs, an exponential relation as suggested by Ullrich et al. (2017) is assumed and has the form


$$n_S = exp(a1 \cdot T + a2)\; m^{-2} \tag{2}$$

, where T is the activation temperature of INPs in K. The INAS density $n_S$ in m$^{-2}$ is calculated by normalizing the measured INP concentration by the total surface area concentration derived from DMPS and APS size distribution data. We adjusted the

parameters in Eq. (2) to our measured INAS densities considering the data set in three different periods corresponding to winter, summer and transition periods. The periods are defined by the measured snow depth s (s = 0 cm corresponds to summertime, s > 10 cm corresponds to wintertime and 0 < s ≤ 10 cm is the transition period.). The new fits yield a1 = -0.543 ± 0.007 K$^{-1}$ and a2 = 154 ± 2 for summertime (R² = 0.78), a1 = -0.495 ± 0.008 K$^{-1}$ and a2 = 141 ± 2 for wintertime (R² = 0.78) and a1 = -0.49 ± 0.01 K$^{-1}$ and a2 = 140 ± 3 for the transition period (R² = 0.85). In Fig. 8b the INP concentration calculated

with these new parameterizations are compared to the measured concentrations. In summertime 92.57% (79.40%, 41.32%) of the data is reproduced by the INAS density fit within a factor of 10 (5, 2), in wintertime 97.32% (86.30%, 47.80%) and in the transition period 99.11% (95.56%, 56.89%) are reproduced.

**4 Conclusions**

This study provides a unique dataset of continuously recorded INP concentrations for more than one year at the SMEARII station located in the Finnish boreal forest. The observations illustrate that the boreal forest is an important source of biogenic INPs with resulting concentrations comparable to other environments (Kanji et al., 2017). We observe a clear seasonal cycle of INP concentrations and INP types in the boreal forest and conclude that this cycle is linked to the prevalence of biogenic aerosol particles. We suggest that these particles are primarily particles emitted by the forest vegetation, but are also correlated with the variable biogenic activities in the forest which appear to contribute both to INPs and to NPF. Current parameterizations do not represent the newly observed seasonal INP variability or concentrations. Thus, we suggest two new approaches for formulating and quantifying the annual variability of INPs over boreal forest areas. The first is a mechanistic approach, which considers the boreal forest as a temperature dependent INP emission source with a pronounced seasonal cycle. The second formulates season specific INAS parameterizations for boreal forest aerosol particles. The new formulations of both approaches reproduce almost all the data points of our long-term record of INP concentrations to within a factor of 10 and provide a basis for models to assess the global or regional importance of boreal forest INPs. As INPs strongly influence precipitation formation and cloud evolution, a description of INPs in weather forecast models is crucial. This study shows that the ice nucleation activity in the atmosphere is highly variable depending on the surrounding conditions. Therefore, it is important to investigate INP concentrations and INP types in different characteristic locations on Earth to establish an overall picture of the global INP abundance and variability. For investigating long-term variability, continuous long-term observations are needed to get a profound insight on the ice nucleation activity at a specific site with a good statistics accounting for the difficulties and uncertainties in INP measurements. With this study, we provide a first step to this overall picture by characterizing the INP population abundant in a remote location in the boreal forest. With continuous aerosol filter sampling for more than one year, we provide the first observation of a clear seasonal cycle, which seems to be dominated by the abundance biogenic aerosol. As in this remote environment, biogenic aerosols seem to play an important role, in other areas, the INP population might be dominated by other species. For further studies, we suggest to conduct further continuous long-term measurements of INPs at different locations on Earth, like anthropogenic influenced locations or deserts. Measurements with a higher time resolution might be useful to investigate relations to meteorological events like precipitation and frontal passages in more detail.

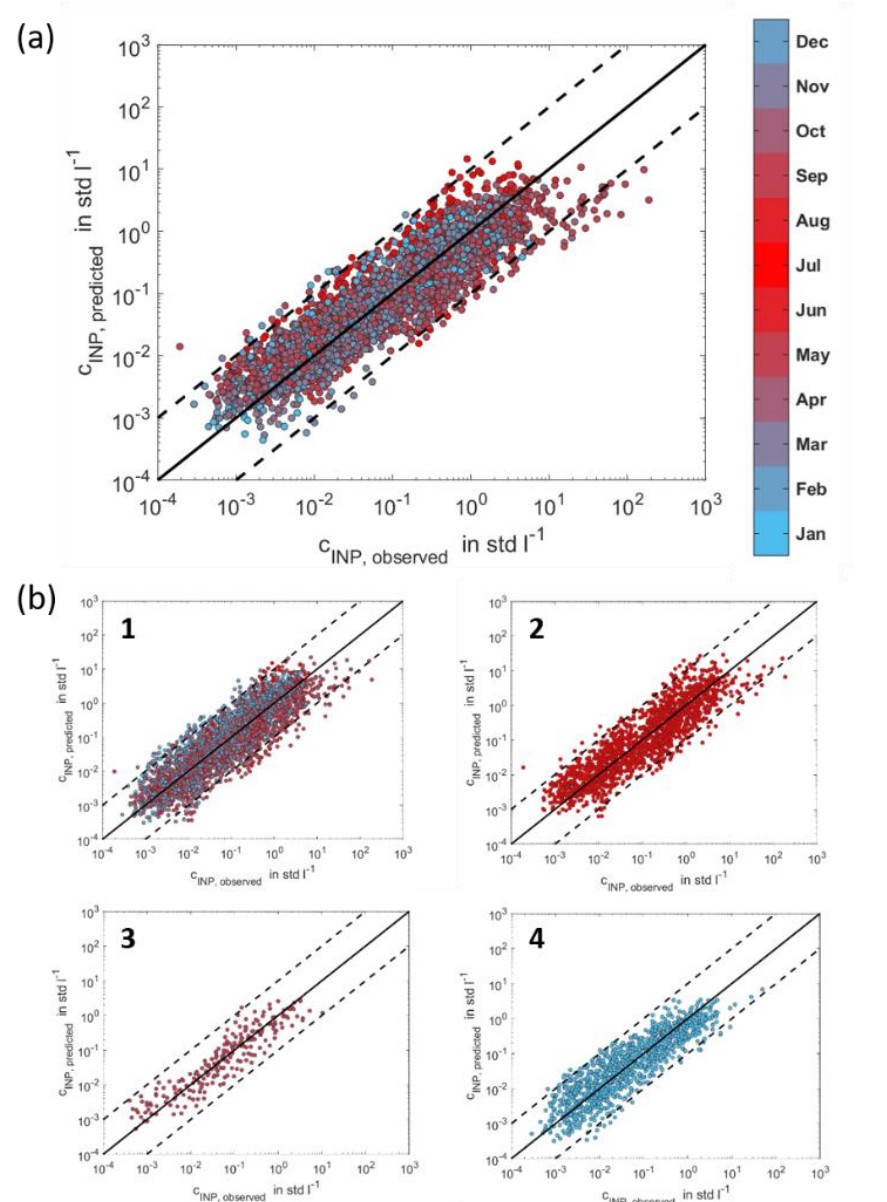


**Figure 8: New INP parameterizations for boreal forest INPs.** Panel (a) shows INP concentrations predicted by our new parametrization using ground-level ambient air temperature compared to measured INP concentrations colour-coded with the corresponding month. In panel (b), the measurements are compared to the predictions of the new INAS density parametrization. The four panels show the new INAS parameterization for the boreal forest INP over the whole year (1), only describing summertime
INPs (2), the parameterization for the transition period (3) and the formulations for boreal INPs in wintertime (4).

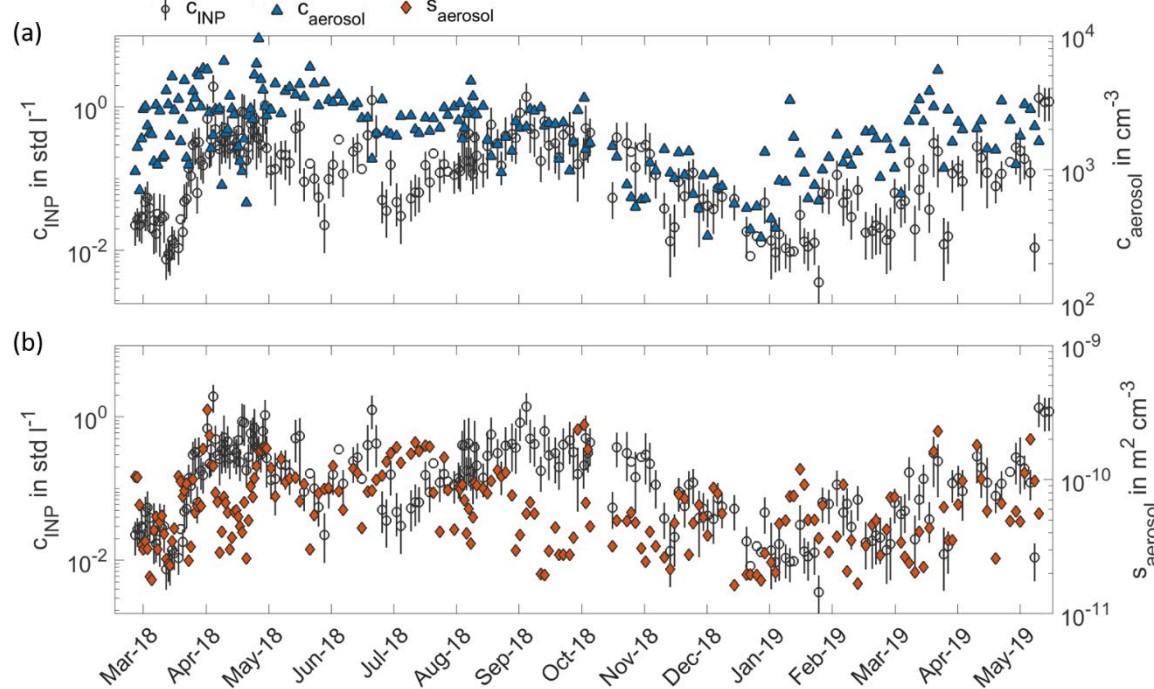

**Figure A1. Time series of number and surface concentration of atmospheric PM10 aerosol. Panel (a) shows the number concentration of atmospheric PM10 aerosol particles measured by DMPS and APS (blue triangles) in comparison to the INP time**
**series at 257 K (circles) from 11 March 2018 to 31 May 2019. In panel (b), the time series of PM10 surface concentrations (orange diamonds) is compared to the INP time series for the same time period.**

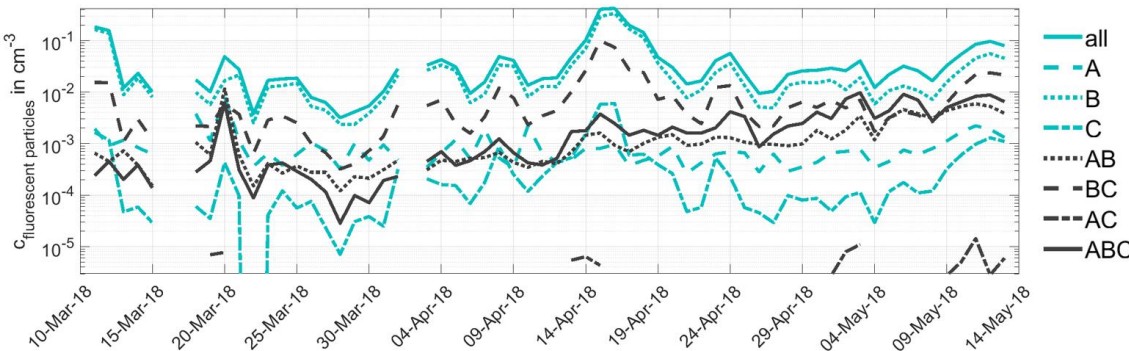

**Figure A2: Time series of fluorescent particle number concentrations. The number concentrations of fluorescent particles measured by WIBS in different excitation emission wavelength pair combinations (fluorescence groups) are shown from 11 March 2018 to 13**
**May 2018. The definition of the different fluorescence groups is based on the categorization given in Savage et al. (2017). A summary of this categorization is given in Section 2.3.**

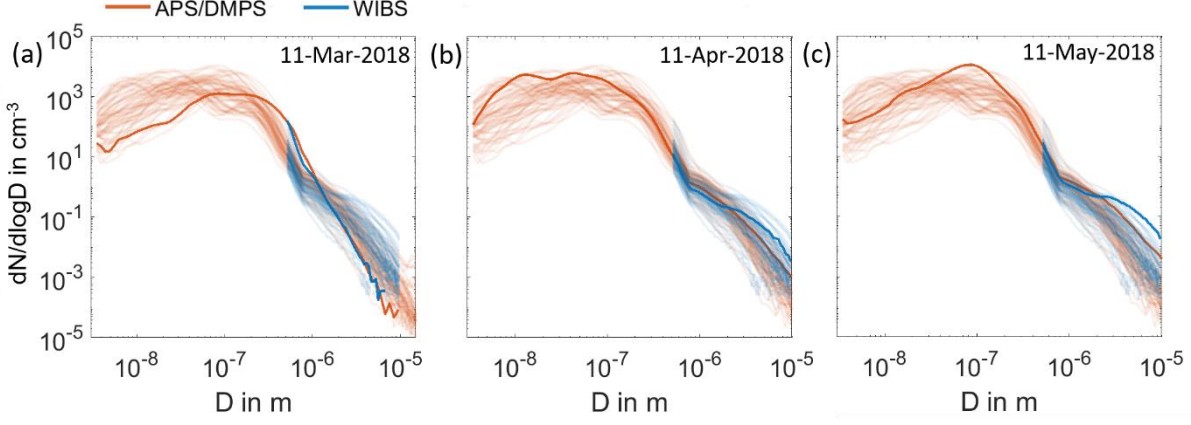

**Figure A3: Aerosol size distributions measured by DMPS/APS and WIBS.** The daily size distributions of atmospheric aerosol particles measured by DMPS and APS (red) are compared to the daily size distribution of atmospheric aerosol particles measured by WIBS (blue) in the period from 11 March 2018 to 13 May 2018. In each of the panels (a), (b), and (c) one size distribution in March, April and May is highlighted.

*Data availability.* The measurement data used in this study is available via KITopen data repository under https://doi.org/10.5445/IR/1000120666 (Schneider et al., 2020).

*Author contributions.* JS wrote this manuscript, did the INP data analysis and part of the filter analysis. KH and OM both strongly supported the analysis and the writing. OM, MK, TP, JD and DM initiated and planned the HyICE-2018 campaign. JS, KH, BB, TS, NSU, FV, PB, PH, ZB, MPA, BJM, EST, DC, KK, YW, DM and JD conducted the measurements within the HyICE-2018 campaign and contributed to the data analysis and the interpretation. PH provided the analysed WIBS data and instrument description. SH provided the analysed NPF event data. LH provided the analysed L-ToF-AMS data and instrument description. JK, TP and TL contributed to the discussion and interpretation of the data and the manuscript writing.

*Competing interests.* The authors declare no competing interests.

*Acknowledgements.* This project received funding from the European Union's Horizon 2020 research and innovation programme under grant agreements No 654109 and 739530 and Transnational access via ACTRIS-2 HyICE-2018 TNA project. The work of the KIT Institute for Meteorology and Climate Research (IMK-AAF) was additionally supported through the Research Programm "Atmosphere and Climate (ATMO)" of the Helmholtz Association and by the KIT Technology Transfer Project PINE (N059). The work of University of Helsinki was supported by the Academy of Finland Centre of Excellence in Atmospheric Science (grant no. 307331) and NANOBIOMASS (307537), ACTRIS-Finland (328616), ACTRIS-CF (329274) and Arctic Community Resilience to Boreal Environmental change: Assessing Risks from fire and disease (ACRoBEAR, 334792) Belmont Forum project. In addition, the work of University of Helsinki was financially supported by European Commission through ACTRIS2 (654109) and ACTRIS-IMP (871115) and ACTRIS2 TransNational Access and through integrative and Comprehensive Understanding on Polar Environments (iCUPE, 689443), ERA-NET-Cofund and by University of Helsinki (ACTRIS-HY). BJM and MPA acknowledge the European Research Council, ERC, MarineIce 648661 for funding. PH and JK acknowledge the funding from the Arctic Academy Programme "ARKTIKO" of Academy of Finland under grant No 286558, and PH from the Maj and Tor Nessling Foundation. EST and DC were supported by the Swedish Research Councils, VR (2013-05153) and FORMAS (2017-00564). NSU acknowledges the support of the Alexander von Humboldt Foundation, Germany (1188375).

Janne Levula, Matti Loponen and rest of the technical staff at the SMEARII station are gratefully acknowledged for their efforts during the HyICE-2018 intensive campaign and during the extended filter sampling period. Support by the IMK-AAF technical team in preparing and operating the INP instruments is also gratefully acknowledged.

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
