# Peer review of "The seasonal cycle of ice-nucleating particles linked to the abundance of biogenic aerosol in boreal forests"

_Atmospheric Chemistry and Physics, 2020_

## Referee Comment (RC1) · Anonymous Referee #1 · 28 Aug 2020

This study shows the seasonal cycle of immersion-mode ice nucleating particles (INPs) in boreal forests in Finland. Since few studies have reported the seasonal cycle of atmospheric INPs in a forested site based on year-round measurements, the INP datasets presented here are unique and will be valuable for atmospheric science communities. This study also suggests that the seasonal cycle of INPs would be attributable to biogenic aerosol particles and provides two new INP parameterizations related to ambient temperature and ice nucleation active site (INAS) approach. However, since evidences to support these points are insufficient, the authors should conduct an in-deep discussion based on other available datasets (e.g., meteorological data, WIBS and L-RoF-AMS data) to quantify the possible contributions of biogenic aerosol particles to INP populations. I would like to support the publication if the authors can answer the following requests/comments.

Major Comments:

1) Further detailed discussion is necessary to investigate a possible relationship between INPs and primary biological aerosol particles (PBAPs). For example, this study uses ambient temperature data as a proxy of changes of the season. As illustrated in Figure 4, it seems that the variation of INPs active at given temperatures are somewhat related to that of ambient temperatures. On the other hand, the comparison with other meteorological parameters is not performed, despite the fact that many earlier studies (e.g., Heald and Spracklen, GRL 2009; Hoose et al., ERL 2010; Huffman et al., ACP 2013; Wright et al., Aerosol Sci. Technol. 2014) have suggested that humidity would be a key factor influencing the release of PBAPs and/or INPs. In addition, as cited in the Introduction section of this paper, rain events might also be an important factor. The authors should conduct further comparison with other meteorological parameters.

2) In addition to the meteorological parameters and snow coverage/depth, the authors might need to check indicators showing the seasonal cycle of vegetation in/around the monitoring site (if available). For example, some earlier studies have used leaf are index (LAI) data as a measure of vegetation density (e.g., Heald and Spracklen, GRL 2009; Hoose et al., ERL 2010).

3) After checking the points described in the above Comments 1 and 2, I would like to suggest reconsidering the INP parameterization related to ambient temperature (equation 1). I admit that this parameterization seems to successfully reproduce the INP number concentrations at this site as compared with other existing parameterizations. However, it is skeptical if this parameterization is scientifically meaningful, while it may be technically useful for reproducing only INP number concentrations near the ground level of this site. Given that this study emphasizes the possible contribution of PBAPs to INP populations, the development of parameterizations related to PBAPs and/or

factors influencing the release of PBAPs would be more scientifically valuable. For example, given that previous model studies have used humidity and LAI to simulate the release of PBAPs, it might be interesting to evaluate if these parameters would also be useful for developing new INP parameterizations.

4) As for the datasets during the HyICE-2018 campaign, the WIBS data would be potentially powerful, because these data might be useful as an indicator of biogenic aerosol particles. Unfortunately, however, the explanations and discussion on these data are insufficient. The authors should add detailed explanations and discussion regarding the WIBS data. For example, the WIBS has multiple channels as illustrated in Figure A2, but I could not find any detailed explanations and discussion about these data in the main text. I assume the categorization, such as FL1, FL2, and FL3, is based on Savage et al. (AMT, 2017). If so, the authors should clarify the source of this definition and then clearly explain the characteristics of each channel provided in Figure A2 (for example, what materials are expected to be detected using FL1, FL2, FL3, and their combinations?). In addition, the authors may need to discuss what channel was most sensitive to the variations of INPs.

5) The explanations on the L-RoF-AMS data are also insufficient. The authors should briefly explain what kinds of organics were measured using the L-RoF-AMS and how they obtained organic mass concentrations presented in Figure 3.

Specific Comments:

6) I could not find any descriptions of the objective of this study in the Introduction section. I thought that most sentences in the first paragraph of the Methods section (Lines 75-88) should be included in the Introduction section.

7) Sections 2.1: What is the top height of the inlet? It is also important to provide the information on vegetation of the SMEARII site (e.g., tree canopy heights, dominant vegetation types) and the difference of the heights between the inlet and canopy.

8) Line 189 (and Figure 3): What is the definition of snow coverage? Where and how was the snow coverage measured?

9) Line 190: The authors would need to briefly explain about how pollen and other PBAPs reported by Manninen et al. (2014) have been measured at the SMEARII site.

10) Line 208 (and Figure 4b): What is the definition of snow depth? Where and how was the snow depth measured?

11) Figure 1: The size of numbers in x-axis and y-axis are too small.

12) Figure 3a: What is the unit of y-axis? As for pollen and PBAP data, since pollen is a kind of PBAPs, I think these should be treated as "pollen" and "other PBAPs". Also, since the pollen and other PBAP data were obtained in 2003/2004, these data should not be included in Figure 3a. I would like to suggest making another figure if they want to show the pollen and other PBAP data in 2003/2004.

13) Figure 3b: What is the unit of y-axis? What is the source of fluorescent particles in this figure? I guessed that it would be the same as "all" in Figure A2 according to a description "all three lase-channel combinations of WIBS" in the main text (lines 205-206). However, these curves in Figures 3b and Figure A2 seems to be quite different. Please clarify this point (see also Comment 4).

14) Figure 3b: I would like to suggest including the data on the number and/or surface area concentrations in PM10 (the data in Figure A1) during the HyICE period in this figure. Then, the authors should discuss if the variations of INPs are indeed related to organics and fluorescent particles, rather than PM10 particles.

15) Figure 6d: How did the authors apply the parameterization by Ullrich et al. (2017)? Did they measure the surface area concentrations of mineral dust particles at this site? Or did they simply apply this parameterization to the measured total aerosol surface area concentrations in PM10? Please clarify this point.

16) Figure A1a: I could not understand how to see this figure. What do the authors

mean by "percentage deviation"? Please clearly explain the meaning of this word.

17) What are the valid temperature ranges of equations 1 and 2? Did you apply these equations to all available INP data or INP data in a specific temperature range when you made Figure 7?

---

## Referee Comment (RC2) · Anonymous Referee #2 · 2 Sep 2020

Schneider et al. present a well-written, succinct manuscript describing results from a full year of INP measurements in a boreal forest region. The study involves assessment of INP biogenic sources in addition to development of a new parameterization for boreal forest INPs. While I find the results and new parameterization valuable, there are a few issues with the manuscript that need to be addressed prior to publication.

While there are indeed very few year-long INP measurements at one location, there are several that report such measurements over an inter-seasonal scale (e.g. Šantl-Temkiv et al., 2019; Stopelli et al., 2015, 2016, and 2017). These studies are worth describing in the introduction. Additionally, it would be useful for the authors to report main findings from previous analogous studies to clearly demonstrate a comparison between those previous and the current results. The introduction is very short and could be beefed up by providing more details on these studies, including their limitations to promote the motivation for the current work.

Šantl-Temkiv, T., Lange, R., Beddows, D., Rauter, U., Pilgaard, S., Dall'Osto, M., Gunde-Cimerman, N., Massling, A., and Wex, H.: Biogenic Sources of Ice Nucleating Particles at the High Arctic Site Villum Research Station, Environ Sci Technol, 53, 10580-10590, 10.1021/acs.est.9b00991, 2019.

Stopelli, E., Conen, F., Morris, C. et al. Ice nucleation active particles are efficiently removed by precipitating clouds. Sci Rep 5, 16433 (2015). https://doi.org/10.1038/srep16433

Stopelli, E., Conen, F., Morris, C. E., Herrmann, E., Henne, S., Steinbacher, M., and Alewell, C.: Predicting abundance and variability of ice nucleating particles in precipitation at the high-altitude observatory Jungfraujoch, Atmos. Chem. Phys., 16, 8341–8351, https://doi.org/10.5194/acp-16-8341-2016, 2016.

Stopelli, E., Conen, F., Guilbaud, C., Zopfi, J., Alewell, C., and Morris, C. E.: Ice nucleators, bacterial cells and Pseudomonas syringae in precipitation at Jungfraujoch, Biogeosciences, 14, 1189–1196, https://doi.org/10.5194/bg-14-1189-2017, 2017.

In regard to the very short introduction, perhaps more time could be spent on: (1) the motivation and objectives of the study itself (i.e. SMEAR II) and (2) more details on current parameterizations and modelling efforts for bioaerosols and biological INPs, which often are conflicting and based on a very limited subset of INP observations. This would inherently provide a clear motivator for developing the boreal INP parameterization.

The snow cover information is useful and corroborates the INP concentration cycle, but what about the transition between melt and full growth of vegetation? Showing some sort of vegetation index and/or type information would be useful, particularly for the inter-seasonal transitions.

The methodology on the WIBS and L-ToF-AMS is incredibly limited. Because data from these methods are presented in the paper, the methods should include sufficient descriptions on each instrument, their operating parameters during SMEAR II, and data analysis and interpretation. Even though the L-ToF-AMS is presented in detail in Paramonov et al. (2020), there should still be a brief description of the instrument and data produced here.

Figure 1: It would be useful to show an averaged spectrum per month overlaid on the data in each panel.

Figure 2: The data in panel a are redundant from Figure 1. Suggest omitting and just keeping panel b.

For the "bulge" which is more pronounced in the heated versus unamended INP spectra for the summer samples, why would this be? There should be some discussion as to why this feature is more prominent when the samples were heated, and why this would occur only for samples collected during the summer.

Because $n_s$ is shown earlier on than page 11, the calculation should be provided in the methods.

Like the introduction, the conclusions are brief and somewhat limited. The "bigger picture" should be reiterated for context of the measurements, and perhaps some discussion on what the authors recommend for the next step and future work.

---

## Referee Comment (RC3) · Anonymous Referee #3 · 10 Sep 2020

In the manuscript titled "The seasonal cycle of ice-nucleating particles linked to the abundance of biogenic aerosol in boreal forests", Schneider et al. describe results from a year-long measurement campaign of ice nucleating particles (INPs) over a forested site in Finland. The data are unique and provide an additional constraint for INPs present over a boreal environment, which are of value to the aerosol-cloud interaction community. I have several comments, most of which I think can be addressed by the authors and I hope to support the publication of this manuscript once these comments have been adequately addressed.

SECTION 1 INTRO The introduction is concise and has room to address two current

missing items: 1) a description of the goal of the manuscript or study; and 2) because one of the main deliverables of this paper is parameterized INP concentrations at this location, it is important to describe the physical mechanisms that INPs may be generated locally or other regional aerosol sources that may impact the INP populations - this is important in creating a physically-based parameterization that may one day be expanded to modeling studies.

SECTION 2 METHODS The methods describing measurements of INPs is quite thorough. I would like the authors to expand on the uncertainty description for INPs - it is not clear where the systematic error percentages were derived. Were blanks collected?

The inlet for the "additional instrumentation" was heated and "RH remains below 40%" – how does this compare to the inlet used for the INP filter collections? Without knowing the exact RH at this site, I suspect RH can get quite high and may increase collection efficiencies of large particles.

For the other measurements, there seems to be a few missing details: - What exactly was measured by the L-ToF-AMS? What sizes are detectable by this measurement? - Was the WIBS connected to a similar inlet as the other instruments? Where large super micron particles measurable? Could the authors include a comparison between the scattering particle size distribution measured from the WIBS and the APS data?

SECTION 3 RESULTS AND DISCUSSION Figure 1 is nearly impossible to read, consider expanding this figure to take up the full page and also increase font sizes for the axes.

L200-204 – are the increases in INPs, organic aerosol, and fluorescent particles statistically significant? I'm not sure I would consider the organic and fluorescent particle concentrations to demonstrate "clear increases".

Figure 3 – It looks as though the INP data are smoothed significantly in Figure 3a. This is especially clear when one compared the subfigure (Figure 3b) with the rest of the

timeline. what kind of curve is fit to the INP data in Figure 3a? If there is no curve, why is it so smoothed? The daily data or monthly averages as points with standard deviations would be a better, more clear, way of visualizing the data.

L208 – This section is title" Comparison to meteorology and aerosol properties", yet only temperature and snow cover are compared. Are there other ambient variables that are correlated, for example relative humidity & winds? It is possible that relative humidity or winds may impact emissions?

L258 – While heating does remove a significant portion of the ice nucleation activity, I think it is important for the authors to also acknowledge that the heat-resistant residual is still associated with significant INP numbers. That is, the INP population is not just heat-labile and may not be entirely biological. I think the way the manuscript is currently written indicates that all INPs are biological, but that is not supported by Figure 5, where there are still significant INPs remaining after heating the samples.

L272 – "... as they do not include seasonal dependencies." – this is not necessarily the case. Seasonal variability in INP abundances is accounted for by being linked to aerosol amount (in these cases n500 or nFBAP), which have seasonal variability.

L267 - I understand the WIBS was used to determine the fluorescent biological aerosol particles (FBAPs); How does this method compare to the UV-APS used in the Tobo et al. (2013) study? Given the number of uncertainties associated with measuring fluorescent measurements, can the authors describe the possible differences between the UV-APS and WIBS and how that would impact the performance of the Tobo et al. (2013) parameterization? Also, how does the Boreal forest in this study differ from that measured in Tobo et al. (2013)?

Figure 6 – I think it would be helpful to mention that only data from the HyICE-2018 period were possible to use in the Tobo et al. (2013) (2) parameterization and therefore that panel has fewer points.

L295 – Should clearly state that this parameterization is based on ground-level ambient temperature, not "ambient temperature". If T is described as ambient temperature, a model will implement this as the temperature predicted at any level, which I do not think is the intention. Additionally, while I understand the impressive correlation between INP and ambient temperature is inviting for a parameterization, I caution the authors in publishing this as an INP parameterization given that it is highly specific to this location and has not been tested for other years, or is not really supported by a physical mechanism. For example, aerosol-based INP parameterizations have a physical mechanism – an aerosol particle that is seasonally variable with an assumed ice nucleation density. The second parameterization presented in this paper has more physical meaning, linking INP abundance to the physical process of snow melting and exposed surface emissions. Without a clear physical basis for a "ground-level temperature-based" parameterization, I would recommend removing this.

L311 – What is T in this equation?

---

## Author Comment (AC1) · 6 Nov 2020

**Response to the interactive comment by Referee #1:**

We thank referee #1 for his or her thoughtful comments and feedback. Please find below our responses and suggestions for the manuscript revision, with the referee comments in black, our answers in red, and suggested changes and/or additions to the manuscript in blue.

This study shows the seasonal cycle of immersion-mode ice nucleating particles (INPs) in boreal forests in Finland. Since few studies have reported the seasonal cycle of atmospheric INPs in a forested site based on year-round measurements, the INP datasets presented here are unique and will be valuable for atmospheric science communities. This study also suggests that the seasonal cycle of INPs would be attributable to biogenic aerosol particles and provides two new INP parameterizations related to ambient temperature and ice nucleation active site (INAS) approach. However, since evidences to support these points are insufficient, the authors should conduct an indeep discussion based on other available datasets (e.g., meteorological data, WIBS and L-RoF-AMS data) to quantify the possible contributions of biogenic aerosol particles to INP populations. I would like to support the publication if the authors can answer the following requests/comments.

Major Comments:

1) Further detailed discussion is necessary to investigate a possible relationship between INPs and primary biological aerosol particles (PBAPs). For example, this study uses ambient temperature data as a proxy of changes of the season. As illustrated in Figure 4, it seems that the variation of INPs active at given temperatures are somewhat related to that of ambient temperatures. On the other hand, the comparison with other meteorological parameters is not performed, despite the fact that many earlier studies (e.g., Heald and Spracklen, GRL 2009; Hoose et al., ERL 2010; Huffman et al., ACP 2013; Wright et al., Aerosol Sci. Technol. 2014) have suggested that humidity would be a key factor influencing the release of PBAPs and/or INPs. In addition, as cited in the Introduction section of this paper, rain events might also be an important factor. The authors should conduct further comparison with other meteorological parameters.

We agree to the referee that correlations to other parameters may be worth considering. In fact we have already explored more than shown in the first manuscript version, but did not find strong correlations. Nevertheless, we have followed the referees suggestion and extended Figure 4 (in the new version, this is Figure 5) by including three additional panels showing the time series of relative humidity (Fig. 5c), wind speed (Fig. 5d) and precipitation (Fig. 5e) and ajusted the description accordingly. We included the following section in the manuscript:

Figure 5c shows the comparison of the INP time series with the time series of relative humidity (RH) measured 35 m above ground. Over the entire time period, no clear relationship between RH and INP concentrations is observed. For a shorter time period from June 2018 to September 2018, there seems to be some correlation of the INP concentration with the RH, during which the peaks in INP concentration in June corresponds to a RH peak. In Figure 5d, we compare the time series of INP concentrations with the time series of wind speed measured 34 m above the ground, which is also above the forest canopy. A relationship between measured INP concentrations and wind speed is not observed. In Figure 5e, the time series of INP concentration is compared to the occurrence of precipitation. Although other studies like Prenni et al. (2013), Huffman et al. (2013) and Iwata et al. (2019) report increasing INP concentrations during and after rain events in forested sites, we do not consistently observe this behavior. In this respect, increased INP concentrations are observed only during two of the strongest precipitation events in June and September 2018. However, it should be noted that in the cited studies INP concentrations were measured with higher time resolutions from minutes to hours. Huffman et al. (2013) reported increased INP and biological particle concentrations during rain events and up to one day after rain events. With our sampling (and therefore averaging)

time of 24 hours or more, rain-induced enhancements of INP concentrations may have been missed. Therefore, this type of sampling strategy may not be appropriate to deterministically link INP concentrations with rain events.

We do not exclude that the INP concentrations measured in the boreal forest are influenced by precipitation events for the whole year. However, our sampling set up is not appropriate to investigate the relationship more precisely.

2) In addition to the meteorological parameters and snow coverage/depth, the authors might need to check indicators showing the seasonal cycle of vegetation in/around the monitoring site (if available). For example, some earlier studies have used leaf are index (LAI) data as a measure of vegetation density (e.g., Heald and Spracklen, GRL 2009; Hoose et al., ERL 2010).

For our measurement site at SMEARII, the vegetation indexes NDVI (Normalized Difference Vegetation Index) and PRI (Photochemical Reflectance Index) are available, but only from 11 March 2018 to 11 September 2018. The NDVI and PRI data and the following interpretation is provided by Jon Atherton and Pasi Kolari from University of Helsinki (personal communication, October, 2020). For both vegetation indices, we calculated daily average values, which are averaged between 12:00 and 13:00 Finnish time, because the solar angle plays an important role for these indices. Unfortunately, LAI is not available.  NDVI mainly tracks the development and loss of green leaf material including overstory trees and understory shrubs. We observe a trend in NDVI over the seasonal transition as the canopy slowly greens from winter to summer. During summer, when the color of the canopy does not change significantly, the NDVI remains relatively constant. The increase in NDVI in the transition period from winter to summer is observed some days later than the increase in INP concentrations. The increase in INP concentrations is nicely represented by increasing air temperature and melting snow in early April 2018 (see Figure 4), whereas a strong increase in NDVI is observed in the second half of April. It has to be considered, that the NDVI is affected by snow resulting in smaller NDVI values. The PRI measures the abundance of photoprotective pigments called carotenoids, which help the tree to deal with excess light, which is potentially harmful. We observed a PRI increase in May, which indicates a carotenoid change of the trees. A relationship to the increasing INP concentrations in April is not observed.

We do not think that these comparisons are of great benefit to the story outlined within the manuscript and have decided not to include the comparison in the paper. However, we are glad it can be made available in this publicly accessible discussion and gratefully acknowledge the support of Jon Atherton and Pasi Kolari by providing and interpreting the NDVI and PRI data.

3) After checking the points described in the above Comments 1 and 2, I would like to suggest reconsidering the INP parameterization related to ambient temperature (equation 1). I admit that this parameterization seems to successfully reproduce the INP number concentrations at this site as compared with other existing parameterizations. However, it is skeptical if this parameterization is scientifically meaningful, while it may be technically useful for reproducing only INP number concentrations near the ground level of this site. Given that this study emphasizes the possible contribution of PBAPs to INP populations, the development of parameterizations related to PBAPs and/or factors influencing the release of PBAPs would be more scientifically valuable. For example, given that previous model studies have used humidity and LAI to simulate the release of PBAPs, it might be interesting to evaluate if these parameters would also be useful for developing new INP parameterizations.

As the time series of vegetation indices available for SMEAR II do not show the same pattern as the time series of INP concentrations, and RH only correlates with INP concentrations in summer months, we do not get a better parameterization of INP concentrations using these parameters. Moreover, ground-level ambient air temperature represents the variability in measured INP concentrations better than other meteorological parameters or the vegetation indices. We are aware, that a physical mechanism that could explain this behavior remains unclear and that this parameterization is only empirical. We report this relationship simply as an observation and provide a description that technically describes INP concentrations quite well in this environment. We also see this result as a motivation for further studies and measurements regarding these findings.

4) As for the datasets during the HyICE-2018 campaign, the WIBS data would be potentially powerful, because these data might be useful as an indicator of biogenic aerosol particles. Unfortunately, however, the explanations and discussion on these data are insufficient. The authors should add detailed explanations and discussion regarding the WIBS data. For example, the WIBS has multiple channels as illustrated in Figure A2, but I could not find any detailed explanations and discussion about these data in the main text. I assume the categorization, such as FL1, FL2, and FL3, is based on Savage et al. (AMT, 2017). If so, the authors should clarify the source of this definition and then clearly explain the characteristics of each channel provided in Figure A2 (for example, what materials are expected to be detected using FL1, FL2, FL3, and their combinations?). In addition, the authors may need to discuss what channel was most sensitive to the variations of INPs.

Thanks for this comment. We agree that a more detailed description of the WIBS instrument and a more thorough discussion of the WIBS data is important to underpin the results of this study. Therefore, we have added a more detailed description of the WIBS instrument to Section 2.3. "Additional Instrumentation at SMEARII", which addresses the question and suggestion about the WIBS and also addresses the associated comments of the other referees.

The WIBS-NEO (Droplet Measurement Technologies, Longmont, CO, USA) is a bioaerosol sensor that provides information on the fluorescence properties, size and asphericity ratio of individual aerosol particles. It operates with an inlet flow of 0.3 l min$^{-1}$ and detects particles with diameters between 500 nm and 30 μm. From 11 March 2018 to 2 April 2018, the WIBS was located about 50m from the aerosol filter sampling line used for the INP analysis. There, it was attached to a total aerosol inlet, which is characterized in Vogel (2018). On 3 April 2018, the WIBS was moved and installed directly next to the filter sampling line and attached to a PM10 inlet, which is described in Schmale et al. (2017). For the WIBS data analysis, particles from 0.5 μm to 10 μm were considered. To analyse the fluorescence of the particles, the WIBS sensor utilizes two xenon flashlamps as excitation light sources (optically filtered at wavelengths of 280 nm and 370 nm) and two emission detection channels (wavelength bands 310 – 400 nm and 420 – 650 nm). Optical size information is acquired utilizing elastic scattering from a continuous wave laser with a wavelength of 635 nm and a photomultiplier tube located orthogonally with respect to the laser. The excitation pulses are fired into the sample volume at different times and both detection channels record the emission(s) from both excitations, leading to three distinguishable excitation-emission combinations (the 370 nm light saturates the 310 – 400 nm detection channel and therefore does not provide any information). Thus, the fluorescence can be divided into 7 unique fluorescence groups based on the excitation-emission wavelength pairs and their combinations, after Perring et al. (2015) and Savage et al. (2017): A (only FL1: excitation 280 nm, emission 310 – 400 nm), B (only FL2: excitation 280 nm, emission 420 – 650 nm), C (only FL3: excitation 370 nm, emission 420 – 650 nm), AB (FL1 + FL2), BC (FL2 + FL3), AC (FL1 + FL3) and ABC (FL1 + FL2 + FL3).

The WIBS performs an empty-chamber background signal check every 8 hours, during which the excitation pulses are fired into the optical chamber without any present particles. The background check collects a multitude of emission intensities that form a baseline for particle fluorescence. In this study, a particle is considered fluorescent, if the associated emission peak intensity is larger than $FT + 9\sigma$. $FT$ is the mean value of the forced trigger intensities and $\sigma$ is their standard deviation.

A more commonly used method would be to compare the emission peak intensity to $FT + 3\sigma$. However, some non-biological particle types such as wood smoke, African dust and black carbon are weakly fluorescent and therefore might satisfy the lower threshold value, leading to an overestimation of biological particle concentration. Furthermore, the stricter threshold only marginally affects the detection efficiency of biological particles, because they tend to have stronger fluorescence (Savage et al., 2017). More detailed descriptions on the WIBS are also available in Savage et al. (2017) and Perring et al. (2015).

We also added the following text to the description on Figure 4 (old Figure 3b) to Section 3.2 Comparison to meteorology and aerosol properties to explain the characteristic of the different excitation-emission wavelengths pairs:

The time series of the number concentration of particles with a fluorescence signal in other fluorescence groups is shown in the Appendix in Fig. A2. In this Figure, the strongest seasonal increase in the transition period from winter to summer is observed in the group ABC. Consequently, this fluorescent group correlates best with the measured INP concentrations (see Figure 4). The characteristics of each fluorescence group are comprehensively investigated and reported in Savage et al. (2017), who examined the fluorescence emissions of different types of pollen, fungi, bacteria, biofluorophores, dust, HULIS (humic-like substances), PAH (polycyclic aromatic hydrocarbons), soot and brown carbon. Using the $FT + 9\sigma$ threshold for defining a particle as fluorescent, nearly all dust and HULIS types show no fluorescence signal at all. Some of the soot and brown carbon types only show weak signals in A, and B, BC and A, respectively. Nearly all of the bacteria types show fluorescence only in group A. The fluorescence of fungal spores are also mainly detected in group A, but also in AB and ABC. The investigated biofluorophores show mainly fluorescence in the groups BC (Riboflavin, NAD), A (Pyridoxamine), AB (Tryptophan) and ABC (Ergosterol). PAHs show fluorescence mostly in groups ABC and A. Finally, most pollen types show fluorescence in groups ABC and AB. Some pollen types also show a fluorescence signal in groups A and B.

We have also modified the discussion on the comparison of WIBS data to INP in the same Section as follows:

Such a correlation is also supported by the peaks of pollen and PBAP concentrations observed in snow-free periods in spring and in autumn, and by the increases of the organic aerosol mass concentration and fluorescent particle numbers of group ABC observed in spring. According to the study of Savage et al. (2017), we assume particles of fluorescence group ABC to be mainly pollen, particles containing PAH or Ergosterol, or fungal spores.

As was suggested by Savage et al. (2017), we have also introduced the categorization of fluorescence groups using A, B, C, AB, BC, AC, ABC instead of FL1, DL2, FL3, FL1+FL2, FL2+FL3, FL1+FL3, FL1+FL2+FL3, as this appeared to be less confusing. We changed the legend of Fig. A2 accordingly.

5) The explanations on the L-RoF-AMS data are also insufficient. The authors should briefly explain what kinds of organics were measured using the L-RoF-AMS and how they obtained organic mass concentrations presented in Figure 3.

Thanks for the comment. We agree that the description and discussion of the L-Tof-AMS needs to be more detailed and precise. We have added a more detailed description of the L-ToF-AMS instrument to Section 2.3. "Additional Instrumentation at SMEARII", which addresses this comment and also the associated comments regarding the L-Tof-AMS description from other referees:

The size-resolved chemical composition of ambient aerosol was measured with the L-ToF-AMS. Its application in the same campaign has been described in Paramonov et al. (2020). It builds on the functionality and characteristics of the high-resolution ToF-AMS (DeCarlo et al., 2006). However, due to the longer time-of-flight chamber, the L-ToF-AMS, has a better resolution (8000 M/ΔM) than the standard ToF-AMS (2000 M/ΔM in V-mode). Detailed descriptions of the instrument, measurements and data processing are available in other publications (Canagaratna et al., 2007; DeCarlo et al., 2006). In general, the L-ToF-AMS measures the size-resolved, non-refractory composition of submicron aerosols, including organic, sulfate, nitrate, ammonium and chloride. The aerodynamic lens has a 100% transmission range of 75-650 nm (in vacuum aerodynamic diameter; Liu et al. (2007)) and focuses particles into a narrow beam that impacts the surface of a porous tungsten vaporizer heated to 600°C, followed by ionization by a 70eV electron source. Ions are detected by a long time-of-flight mass analyzer (Tofwerk AG). The sample flow of 0.09 l min$^{-1}$ is extracted from an extra suction flow (3 l min$^{-1}$) that is used to avoid aerosol losses in the inlet line. A PM2.5 cyclone mounted at the inlet removes large particles to avoid clogging the critical orifice (100μm), and before entering the L-ToF-AMS, the samples are dried by a Nafion dryer to keep the RH below 30%.

The L-ToF-AMS data were analyzed using standard ToF-AMS data analysis toolkits (Squirrel V1.61B and PIKA1.21B) using Igor Pro software (V6.37, WaveMetrics Inc.). To calculate mass concentrations an ionization efficiency (IE) was determined using 300 nm, size-selected, dry ammonium nitrate particles, and a relative ionization efficiency (RIE) for ammonium of 3.7 was determined. The default relative ionization efficiency (RIE) values of 1.1, 1.2, 1.3 and 1.4 for nitrate, sulfate, chloride and organics, respectively, were applied. A composition-dependent collection efficiency (CE) was applied based on the principles proposed by Middlebrook et al. (2012).

We have also added the following text to the discussion of Figure 4 (old Figure 3b) in Section 3.2 "Comparison to meteorology and aerosol properties."

The non-refractory organic components measured by the AMS include the commonly observed primary organic aerosol (POA) and oxygenated organic aerosol (OOA).

Specific Comments:

6) I could not find any descriptions of the objective of this study in the Introduction section. I thought that most sentences in the first paragraph of the Methods section (Lines 75-88) should be included in the Introduction section.

We agree that the manuscript would benefit from a stronger definition of the objectives of this study – thanks for this comment. We add the following text to the introduction:

The main objective of this study is to investigate and describe the variability and seasonal trends in INP concentrations and INP temperature spectra in a boreal forest environment. The absence of anthropogenic and/or dust aerosol sources in the boreal region motivates the additional

investigation of biogenic ice nucleation activity and reveals the relevance of boreal forest areas as an important INP source. The comprehensive instrumentation provided at the measurement site at the SMEARII station allows comparisons between INP measurements with simultaneous measurements of many meteorological variables. These measurements are complemented by measurements characterizing the sampled aerosol number concentrations, size distributions and chemical compositions in order to elucidate the potential origin and nature of the INPs. Heat treatments of the suspensions prior to INP analysis also help identifying the nature of INPs. We aim to improve the parameterizations describing atmospheric INP concentrations in the boreal forest by considering seasonal dependences in the formulations. Finally, this study provides motivation for further continuous long-term studies of INP in different environments across the globe.

In addition, we have shifted the description of the SMEARII station from the Methods to the Introduction, as the referee suggested.

7) Sections 2.1: What is the top height of the inlet? It is also important to provide the information on vegetation of the SMEARII site (e.g., tree canopy heights, dominant vegetation types) and the difference of the heights between the inlet and canopy.

For the top height of the inlet of the aerosol sampling line, we added the following sentence to Section 2.1 "Aerosol filter sampling":

The inlet height is approximately 4.6 m above ground and therefore approximately 17.2 m below the forest canopy.

Concerning the dominant vegetation species and canopy height, we have added the following information to the introduction into the description of the SMEARII station:

The boreal forest around the SMEARII station is dominated by Scots pine trees (Hari and Kulmala, 2005). In summer 2018, the canopy height of pines at SMEARII was determined to be 21.8 m.

8) Line 189 (and Figure 3): What is the definition of snow coverage? Where and how was the snow coverage measured?

We have added the following explanation to Section 3.2 "Comparison to meteorology and aerosol properties" to explain the definition of snow coverage applied here:

We have defined snow coverage as measured snow depth > 1 cm, where snow depth was measured by a Jenoptik SHM30 snow depth sensor, which is based on an opto-electronic laser distance sensor, in open field about 500 m southeast of the aerosol collection area of SMEARII.

9) Line 190: The authors would need to briefly explain about how pollen and other PBAPs reported by Manninen et al. (2014) have been measured at the SMEARII site.

We have added the following explanation to Section 3.2 "Comparison to meteorology and aerosol properties" to explain the measurements of PBAP in Manninen et al, 2014:

Manninen et al. (2014) collected aerosol samples in a Hirst-type volumetric spore trap (Burkard Manufacturing Co. Ltd.; Hirst, 1952), located at SMEARII 3 m above the forest canopy. The trap is driven by a clockwork and collects aerosol particles larger than approximately 3 μm on an adhesive, transparent, plastic tape with a sampling flow rate of approximately 10 l min$^{-1}$. The analysis of the collected particles was performed according to standard methodology adopted by the Finnish pollen

information network and following the principles of the European Aeroallergen Network (www.polleninfo.org/) and Rantio-Lehtimäki et al. (1994).

10) Line 208 (and Figure 4b): What is the definition of snow depth? Where and how was the snow depth measured?

See answer to Comment 8.

11) Figure 1: The size of numbers in x-axis and y-axis are too small.

We have increased the size of the axis description in Figure 1.

12) Figure 3a: What is the unit of y-axis? As for pollen and PBAP data, since pollen is a kind of PBAPs, I think these should be treated as "pollen" and "other PBAPs". Also, since the pollen and other PBAP data were obtained in 2003/2004, these data should not be included in Figure 3a. I would like to suggest making another figure if they want to show the pollen and other PBAP data in 2003/2004.

We thank the referee for this suggestion. The units for PBAP and INP concentrations has been added ($m^{-2}$ and std $l^{-1}$, respectively). The fraction of NPF event days has no unit. The legend was changed from "PBAP" to "other PBAP". To improve the comparison of data measured in different years, the x-axis labels were adjusted by removing the year(s) and instead the corresponding years are shown in the legend. Now, the presentation of pollen and PBAP data from 2003/2004 together with INP, NPF and snow data from 2018-2019 should be improved. We think it is useful to have this data in one panel, as we focus on general seasonal trends and not on quantitative values measured in specific years. Other improvements to this Figure has been done considering other referee comments.

13) Figure 3b: What is the unit of y-axis? What is the source of fluorescent particles in this figure? I guessed that it would be the same as "all" in Figure A2 according to a description "all three lase-channel combinations of WIBS" in the main text (lines 205- 206). However, these curves in Figures 3b and Figure A2 seems to be quite different. Please clarify this point (see also Comment 4).

We did not use "all", but "FL1+FL2+FL3" (group ABC), which agrees with the curve in Figure A2. We have now completely modified the Figure. For details see answer to comment 14) below. A complete comparison, including aerosol number concentration and surface concentration is too much for a single panel, and thus we prepared a separate Figure including several panels showing INP concentrations compared to ABC fluorescent particle concentration, mass concentrations of organics, PM10 number concentration and PM10 surface concentration (see new Figure 4). We have added corresponding units to the axis labels, where they were missing.

14) Figure 3b: I would like to suggest including the data on the number and/or surface area concentrations in PM10 (the data in Figure A1) during the HyICE period in this figure. Then, the authors should discuss if the variations of INPs are indeed related to organics and fluorescent particles, rather than PM10 particles.

Thanks for this suggestion. Continuing from above in the new Figure 4 panel (a) shows a comparison of INP concentrations to ABC fluorescent particles, panel (b) to organic mass concentrations, panel (c) to PM10 number concentrations and panel (d) to PM10 surface concentrations.
We have added additional description to Section 3.2. "Comparison to meteorology and aerosol properties:"

Both, the organic aerosol mass concentration measured by the L-ToF-AMS and the number concentration of fluorescent particles measured by WIBS, tend to increase during the transition

period from winter to summer (Fig. 4a and b). The number concentration of atmospheric PM10 aerosol presented in Fig. 4c does not show this trend. However, a slight seasonal trend is visible in the PM10 surface concentration (Fig. 4d).

We have also added a short discussion of the previous observation later to the same Section:

As the surface concentration of PM10 particles is more sensitive to larger particles, the increase in PM10 surface concentration (see Fig. 4d) indicates that the observed seasonal increase may be due to larger particles, which are expected to be mainly of biogenic origin.

15) Figure 6d: How did the authors apply the parameterization by Ullrich et al. (2017)? Did they measure the surface area concentrations of mineral dust particles at this site? Or did they simply apply this parameterization to the measured total aerosol surface area concentrations in PM10? Please clarify this point.

The parameterization of Ullrich et al. (2017) for the INAS density of mineral dust is only temperature dependent. Therefore, for the application of this parameterization, we only used the activation temperatures in the Ullrich et al. (2017) formulation. When we established a new parameterization for the INAS densities of the measured boreal forest INPs (see Eq. (2)), we used the same exponential relation of INAS density and activation temperature as suggested in Ullrich et al. (2017). For calculating the INAS densities used to establish this new parameterization, the PM10 surface area concentrations measured by APS/DMPS including all atmospheric aerosol particles were used.

16) Figure A1a: I could not understand how to see this figure. What do the authors mean by "percentage deviation"? Please clearly explain the meaning of this word.

Thanks for this comment. The old Figure A1a was meant to explain and show the difference in percent between the number/surface concentration of total aerosol and PM10 aerosol. As the Figure seems to be more confusing, we decided to remove this Figure and just describe in the text that the difference between total aerosol and PM10 aerosol is lower than 1% (line 151, revised manuscript). The old Figure A1b was separated into two panels and compared to INP concentration at 257 K in the same way as meteorological parameters are compared to INPs and are now the new Figure A1.

We referred to the new Figure in Section 3.2 "Comparison to meteorology and aerosol properties," as follows:

The number and surface concentration of PM10 atmospheric aerosol for the entire time period from March 2018 to May 2019 is shown in the Appendix in Fig. A1.

17) What are the valid temperature ranges of equations 1 and 2? Did you apply these equations to all available INP data or INP data in a specific temperature range when you made Figure 7?

Regarding equation 1, it is mentioned in line 494 that INP-T-spectra between 250-265 K are used. Therefore, the parameterization is valid for this temperature range. Also, for the INAS density parameterization in equation 2, the spectra in the full T range (250-265 K ) was used and is therefore valid in the same temperature range.

**References:**

[revised manuscript text omitted]

---

## Author Comment (AC2) · 6 Nov 2020

**Response to the interactive comment by Referee #2:**

We thank referee #2 for his or her thoughtful comments and feedback. Please find below our responses and suggestions for the manuscript revision, with the referee comments in black, our answers in red, and suggested changes or additions to the manuscript in blue.

Schneider et al. present a well-written, succinct manuscript describing results from a full year of INP measurements in a boreal forest region. The study involves assessment of INP biogenic sources in addition to development of a new parameterization for boreal forest INPs. While I find the results and new parameterization valuable, there are a few issues with the manuscript that need to be addressed prior to publication.

While there are indeed very few year-long INP measurements at one location, there are several that report such measurements over an inter-seasonal scale (e.g. Šantl-Temkiv et al., 2019; Stopelli et al., 2015, 2016, and 2017). These studies are worth describing in the introduction. Additionally, it would be useful for the authors to report main findings from previous analogous studies to clearly demonstrate a comparison between those previous and the current results. The introduction is very short and could be beefed up by providing more details on these studies, including their limitations to promote the motivation for the current work.

Šantl-Temkiv, T., Lange, R., Beddows, D., Rauter, U., Pilgaard, S., Dall'Osto, M., Gunde-Cimerman, N., Massling, A., and Wex, H.: Biogenic Sources of Ice Nucleating Particles at the High Arctic Site Villum Research Station, Environ Sci Technol, 53, 10580-10590, 10.1021/acs.est.9b00991, 2019.
Stopelli, E., Conen, F., Morris, C. et al. Ice nucleation active particles are efficiently removed by precipitating clouds. Sci Rep 5, 16433 (2015). https://doi.org/10.1038/srep16433
Stopelli, E., Conen, F., Morris, C. E., Herrmann, E., Henne, S., Steinbacher, M., and Alewell, C.: Predicting abundance and variability of ice nucleating particles in precipitation at the high-altitude observatory Jungfraujoch, Atmos. Chem. Phys., 16, 8341–8351, https://doi.org/10.5194/acp-16-8341-2016, 2016.
Stopelli, E., Conen, F., Guilbaud, C., Zopfi, J., Alewell, C., and Morris, C. E.: Ice nucleators, bacterial cells and Pseudomonas syringae in precipitation at Jungfraujoch, Biogeosciences, 14, 1189–1196, https://doi.org/10.5194/bg-14-1189-2017, 2017.

Thanks for this suggestion. We agree that the introduction should be more comprehensive and precise. Therefore, we described previous studies, which could be compared to our study, in more detail within the introduction focusing on the presented seasonal cycle, the considered time period, the continuity of measurements and the measurement locations. We also included the suggested publications Santl-Temkiv et al. (2019) and Stopelli et al. (2015, 2016), as those also show seasonal trends in their INP data sets. Stopelli et al. (2017) and Stopelli et al. (2016 ) have been added as references when discussing the relation between INP concentrations and precipitation (lines 44-45, revised manuscript).

We have included the following section to the introduction:

[revised manuscript text omitted]

In regard to the very short introduction, perhaps more time could be spent on: (1) the motivation and objectives of the study itself (i.e. SMEAR II) and (2) more details on current parameterizations and modelling efforts for bioaerosols and biological INPs, which often are conflicting and based on a very limited subset of INP observations. This would inherently provide a clear motivator for developing the boreal INP parameterization.

Explanations of the objectives and motivation for this study have been added to the introduction. We have also extended the discussion of studies, which report about biological INPs and bioaerosols and included a discussion of the difficulties related to investigating and parameterizing these particles.

Text added to the introduction describing motivation:

The main objective of this study is to investigate and describe the variability and seasonal trends in INP concentrations and INP temperature spectra in a boreal forest environment. The absence of anthropogenic and/or dust aerosol sources in the boreal region motivates the additional investigation of biogenic ice nucleation activity and reveals the relevance of boreal forest areas as an important INP source. The comprehensive instrumentation provided at the measurement site at the

SMEARII station allows comparisons between INP measurements with simultaneous measurements of many meteorological variables. These measurements are complemented by measurements characterizing the sampled aerosol number concentrations, size distributions and chemical compositions in order to elucidate the potential origin and nature of the INPs. Heat treatments of the suspensions prior to INP analysis also help identifying the nature of INPs. We aim to improve the parameterizations describing atmospheric INP concentrations in the boreal forest by considering seasonal dependences in the formulations. Finally, this study provides motivation for further continuous long-term studies of INP in different environments across the globe.

Text added to the introduction discussing studies on bioaerosols and biological INP:

Several biogenic aerosol types have been shown to have atmospherically relevant ice-nucleating abilities (Augustin et al., 2013; Creamean et al., 2013; Hader et al., 2014; Möhler et al., 2007; Morris et al., 2004; O'Sullivan et al., 2015, 2018; Pratt et al., 2009; Schnell and Vali, 1973) especially at temperatures above -15°C (Christner et al., 2008; Murray et al., 2012).  Although the contribution of biogenic INPs to the total global INP abundance is thought to be rather low (Hoose et al., 2010), biogenic aerosol may contribute substantially at regional scales where biological aerosol sources are important. For example, Tobo et al. (2013), Prenni et al. (2009) and O'Sullivan et al. (2018) have observed biogenic aerosol in the INP populations of the forested environments in Colorado, in the Amazon basin and in rural areas in Northern Europe. Furthermore, Pratt et al. (2009) and Creamean et al. (2013) showed that biological particles were frequently present in ice crystal and precipitation residues measured over the western United States and suggested that these particles play a key role in cloud ice formation. In a study about Swedish and Czech birch pollen, Augustin et al. (2013) reported the ice-nucleation activity of sampled macromolecules and formulated new parameterizations for the heterogeneous nucleation rates of two different ice-active macromolecules. However, in general, measuring and parameterizing the IN ability of biogenic particles has proven to be difficult for several reasons. For accurate biogenic INP model simulations, it is critical to understand the global distribution of biogenic INP, their source strength and their aerosolization and atmospheric transport mechanisms (O'Sullivan et al., 2018). It remains unresolved how the microphysical and chemical properties of biogenic aerosol may change during transport processes in the atmosphere. In field studies, which attempt to address these deficiencies, it remains difficult to identify biogenic aerosol particles and to separate them from non-biogenic particles (Möhler et al., 2007). Moreover, there are many biogenic species with a range of properties, which complicate comparisons and generalized parameterizations.

The snow cover information is useful and corroborates the INP concentration cycle, but what about the transition between melt and full growth of vegetation? Showing some sort of vegetation index and/or type information would be useful, particularly for the inter-seasonal transitions.

For our measurement site at SMEARII, the vegetation indexes NDVI (Normalized Difference Vegetation Index) and PRI (Photochemical Reflectance Index) are available, but only from 11 March 2018 to 11 September 2018. The NDVI and PRI data and the following interpretation is provided by Jon Atherton and Pasi Kolari from University of Helsinki (personal communication, October, 2020). For both vegetation indices, we calculated daily average values, which are averaged between 12:00 and 13:00 Finnish time, because the solar angle plays an important role for these indices. NDVI mainly tracks the development and loss of green leaf material including overstory trees and understory shrubs. We observe a trend in NDVI over the seasonal transition as the canopy slowly

greens from winter to summer. During summer, when the color of the canopy does not change significantly, the NDVI remains relatively constant. The increase in NDVI in the transition period from winter to summer is observed some days later than the increase in INP concentrations. The increase in INP concentrations is nicely represented by increasing air temperature and melting snow in early April 2018 (see Figure 4), whereas a strong increase in NDVI is observed in the second half of April. It has to be considered, that the NDVI is affected by snow resulting in smaller NDVI values. The PRI measures the abundance of photoprotective pigments called carotenoids, which help the tree to deal with excess light, which is potentially harmful. We observed a PRI increase in May, which indicates a carotenoid change of the trees. A relationship to the increasing INP concentrations in April is not observed.

We do not think that these comparisons are of great benefit to the story outlined within the manuscript and have decided not to include the comparison in the paper. However, we are glad it can be made available in this publicly accessible discussion and gratefully acknowledge the support of Jon Atherton and Pasi Kolari by providing and interpreting the NDVI and PRI data.

The methodology on the WIBS and L-ToF-AMS is incredibly limited. Because data from these methods are presented in the paper, the methods should include sufficient descriptions on each instrument, their operating parameters during SMEAR II, and data analysis and interpretation. Even though the L-ToF-AMS is presented in detail in Paramonov et al. (2020), there should still be a brief description of the instrument and data produced here.

Thanks for this comment. We agree that a more detailed explanation of these measurements is needed to support the results of this study. We have added a more detailed description of the WIBS instrument to Section 2.3. "Additional Instrumentation at SMEARII". As the other referees also asked further questions about the details on the WIBS and L-ToF-AMS, the added description addresses the comments of all the referees regarding this topic:

The WIBS-NEO (Droplet Measurement Technologies, Longmont, CO, USA) is a bioaerosol sensor that provides information on the fluorescence properties, size and asphericity ratio of individual aerosol particles. It operates with an inlet flow of 0.3 l min$^{-1}$ and detects particles with diameters between 500 nm and 30 μm.  From 11 March 2018 to 2 April 2018, the WIBS was located about 50m from the aerosol filter sampling line used for the INP analysis. There, it was attached to a total aerosol inlet, which is characterized in Vogel (2018). On 3 April 2018, the WIBS was moved and installed directly next to the filter sampling line and attached to a PM10 inlet, which is described in Schmale et al. (2017). For the WIBS data analysis, particles from 0.5 μm to 10 μm were considered. To analyse the fluorescence of the particles, the WIBS sensor utilizes two xenon flashlamps as excitation light sources (optically filtered at wavelengths of 280 nm and 370 nm) and two emission detection channels (wavelength bands 310 – 400 nm and 420 – 650 nm). Optical size information is acquired utilizing elastic scattering from a continuous wave laser with a wavelength of 635 nm and a photomultiplier tube located orthogonally with respect to the laser. The excitation pulses are fired into the sample volume at different times and both detection channels record the emission(s) from both excitations, leading to three distinguishable excitation-emission combinations (the 370 nm light saturates the 310 – 400 nm detection channel and therefore does not provide any information). Thus, the fluorescence can be divided into 7 unique fluorescence groups based on the excitation-emission wavelength pairs and their combinations, after Perring et al. (2015) and Savage et al. (2017): A (only FL1: excitation 280 nm, emission 310 – 400 nm), B (only FL2: excitation 280 nm, emission 420 – 650 nm), C (only FL3: excitation 370 nm, emission 420 – 650 nm), AB (FL1 + FL2), BC (FL2 + FL3), AC (FL1 + FL3) and ABC (FL1 + FL2 + FL3).

The WIBS performs an empty-chamber background signal check every 8 hours, during which the excitation pulses are fired into the optical chamber without any present particles. The background check collects a multitude of emission intensities that form a baseline for particle fluorescence. In this study, a particle is considered fluorescent, if the associated emission peak intensity is larger than $FT + 9\sigma$. $FT$ is the mean value of the forced trigger intensities and $\sigma$ is their standard deviation.

A more commonly used method would be to compare the emission peak intensity to $FT + 3\sigma$. However, some non-biological particle types such as wood smoke, African dust and black carbon are weakly fluorescent and therefore might satisfy the lower threshold value, leading to an overestimation of biological particle concentration. Furthermore, the stricter threshold only marginally affects the detection efficiency of biological particles, because they tend to have stronger fluorescence (Savage et al., 2017). More detailed descriptions on the WIBS are also available in Savage et al. (2017) and Perring et al. (2015).

We also added the following text to the description on Figure 4 (old Figure 3b) to Section 3.2 "Comparison to meteorology and aerosol properties" to explain the characteristic of the different excitation-emission wavelengths pairs:

The time series of the number concentration of particles with a fluorescence signal in other fluorescence groups is shown in the Appendix in Fig. A2. In this Figure, the strongest seasonal increase in the transition period from winter to summer is observed in the group ABC. Consequently, this fluorescent group correlates best with the measured INP concentrations (see Figure 4). The characteristics of each fluorescence group are comprehensively investigated and reported in Savage et al. (2017), who examined the fluorescence emissions of different types of pollen, fungi, bacteria, biofluorophores, dust, HULIS (humic-like substances), PAH (polycyclic aromatic hydrocarbons), soot and brown carbon. Using the $FT + 9\sigma$ threshold for defining a particle as fluorescent, nearly all dust and HULIS types show no fluorescence signal at all. Some of the soot and brown carbon types only show weak signals in A, and B, BC and A, respectively. Nearly all of the bacteria types show fluorescence only in group A. The fluorescence of fungal spores are also mainly detected in group A, but also in AB and ABC. The investigated biofluorophores show mainly fluorescence in the groups BC (Riboflavin, NAD), A (Pyridoxamine), AB (Tryptophan) and ABC (Ergosterol). PAHs show fluorescence mostly in groups ABC and A. Finally, most pollen types show fluorescence in groups ABC and AB. Some pollen types also show a fluorescence signal in groups A and B.

We also adjusted the discussion on the comparison of WIBS data to INP in the same Section as follows:

Such a correlation is also supported by the peaks of pollen and PBAP concentrations observed in snow-free periods in spring and in autumn, and by the increases of the organic aerosol mass concentration and fluorescent particle numbers of group ABC observed in spring. According to the study of Savage et al. (2017), we assume particles of fluorescence group ABC to be mainly pollen, particles containing PAH or Ergosterol, or fungal spores.

We also added a more detailed description on the L-ToF-AMS instrument to Section 2.3." Additional Instrumentation at SMEARII":

The size-resolved chemical composition of ambient aerosol was measured with the L-ToF-AMS. Its application in the same campaign has been described in Paramonov et al. (2020). It builds on the functionality and characteristics of the high-resolution ToF-AMS (DeCarlo et al., 2006). However, due to the longer time-of-flight chamber, the L-ToF-AMS, has a better resolution (8000 M/ΔM) than the

standard ToF-AMS (2000 M/ΔM in V-mode). Detailed descriptions of the instrument, measurements and data processing are available in other publications (Canagaratna et al., 2007; DeCarlo et al., 2006). In general, the L-ToF-AMS measures the size-resolved, non-refractory composition of submicron aerosols, including organic, sulfate, nitrate, ammonium and chloride. The aerodynamic lens has a 100% transmission range of 75-650 nm (in vacuum aerodynamic diameter; Liu et al. (2007)) and focuses particles into a narrow beam that impacts the surface of a porous tungsten vaporizer heated to 600°C, followed by ionization by a 70eV electron source. Ions are detected by a long time-of-flight mass analyzer (Tofwerk AG). The sample flow of 0.09 l min$^{-1}$ is extracted from an extra suction flow (3 l min$^{-1}$) that is used to avoid aerosol losses in the inlet line. A PM2.5 cyclone mounted at the inlet removes large particles to avoid clogging the critical orifice (100μm), and before entering the L-ToF-AMS, the samples are dried by a Nafion dryer to keep the RH below 30%.

The L-ToF-AMS data were analyzed using standard ToF-AMS data analysis toolkits (Squirrel V1.61B and PIKA1.21B) using Igor Pro software (V6.37, WaveMetrics Inc.). To calculate mass concentrations an ionization efficiency (IE) was determined using 300 nm, size-selected, dry ammonium nitrate particles, and a relative ionization efficiency (RIE) for ammonium of 3.7 was determined. The default relative ionization efficiency (RIE) values of 1.1, 1.2, 1.3 and 1.4 for nitrate, sulfate, chloride and organics, respectively, were applied. A composition-dependent collection efficiency (CE) was applied based on the principles proposed by Middlebrook et al. (2012).

We also added the following text to the discussion on Figure 4 (old Figure 3b) to Section 3.2 "Comparison to meteorology and aerosol properties" to give more information about the measured organics:

The non-refractory organic components measured by the AMS include the commonly observed primary organic aerosol (POA) and oxygenated organic aerosol (OOA).

Figure 1: It would be useful to show an averaged spectrum per month overlaid on the data in each panel.

With monthly averaging, the variability of the spectra within one month and also the characteristic spectral shape is no longer visible. As this Figure is meant to show the daily/monthly variability and characteristic concentrations and spectral shapes throughout the full year of measurements, we have decided to keep the Figure as is.

Figure 2: The data in panel a are redundant from Figure 1. Suggest omitting and just keeping panel b.

We think it is reasonable to keep panel (a) in Figure 2, as the representation on a time axis allows easier peak identification in the INP concentrations and highlights the seasonal variability and trends.

For the "bulge" which is more pronounced in the heated versus unamended INP spectra for the summer samples, why would this be? There should be some discussion as to why this feature is more prominent when the samples were heated, and why this would occur only for samples collected during the summer.

This is a very interesting point. So far, the "bulge" is only an observation, for which we do not yet have a concrete explanation. We want to show what we observed, which may also be a basis for further studies and measurement campaigns that could directly be initiated with the objective to explain this observation.

Because $n_s$ is shown earlier on than page 11, the calculation should be provided in the methods.

Thanks for this comment. We have added the following section to the Methods:

INAS densities were calculated as described in Eq. (2) in Ullrich et al. (2017), where ice number concentrations are normalized by the aerosol surface area concentration. Assuming that every INP triggers the formation of one ice crystal, the ice number concentrations are equal to the INP concentrations, which are determined by the INSEKT measurements. The aerosol surface area concentrations are derived from continuous size distribution measurements of the PM10 atmospheric aerosol at SMEARII. Details on the size distribution measurements are given in the following section.

Like the introduction, the conclusions are brief and somewhat limited. The "bigger picture" should be reiterated for context of the measurements, and perhaps some discussion on what the authors recommend for the next step and future work.

We agree that the conclusion needed improvements – thanks for this comment. We now set our study in the context of the "bigger picture" and have added suggestions for further work:

[revised manuscript text omitted]

---

## Author Comment (AC3) · 6 Nov 2020

**Response to the interactive comment by Referee #3:**

We thank referee #3 for his or her thoughtful comments and feedback. Please find below our responses and suggestions for the manuscript revision, with the referee comments in black, our answers in red, and suggested changes or additions to the manuscript in blue.

In the manuscript titled "The seasonal cycle of ice-nucleating particles linked to the abundance of biogenic aerosol in boreal forests", Schneider et al. describe results from a year-long measurement campaign of ice nucleating particles (INPs) over a forested site in Finland. The data are unique and provide an additional constraint for INPs present over a boreal environment, which are of value to the aerosol-cloud interaction community. I have several comments, most of which I think can be addressed by the authors and I hope to support the publication of this manuscript once these comments have been adequately addressed.

SECTION 1 INTRO The introduction is concise and has room to address two current missing items: 1) a description of the goal of the manuscript or study; and 2) because one of the main deliverables of this paper is parameterized INP concentrations at this location, it is important to describe the physical mechanisms that INPs may be generated locally or other regional aerosol sources that may impact the INP populations - this is important in creating a physically-based parameterization that may one day be expanded to modeling studies.

We agree with the referee that the paper would benefit from a stronger definition of the objective of the study. Therefore, we have added a description of the objectives of this study at the end of the introduction:

The main objective of this study is to investigate and describe the variability and seasonal trends in INP concentrations and INP temperature spectra in a boreal forest environment. The absence of anthropogenic and/or dust aerosol sources in the boreal region motivates the additional investigation of biogenic ice nucleation activity and reveals the relevance of boreal forest areas as an important INP source. The comprehensive instrumentation provided at the measurement site at the SMEARII station allows comparisons between INP measurements with simultaneous measurements of many meteorological variables. These measurements are complemented by measurements characterizing the sampled aerosol number concentrations, size distributions and chemical compositions in order to elucidate the potential origin and nature of the INPs. Heat treatments of the suspensions prior to INP analysis also help identifying the nature of INPs. We aim to improve the parameterizations describing atmospheric INP concentrations in the boreal forest by considering seasonal dependences in the formulations. Finally, this study provides motivation for further continuous long-term studies of INP in different environments across the globe.

General information about the characteristic aerosol particle population is already given in the introduction. Here, we state that boreal forests are generally far from anthropogenic or dust sources, which consequently are not expected to contribute to the generation of boreal INPs. As the main source of aerosol particle in boreal forest areas appeared to be the forest itself, it is also likely, that the INP populations are mainly influenced and generated by this source. As we present indication for abundant biogenic particles in the INP population, this expectation is underlined. We cannot give more precise description on INP generation processes in the boreal forest, as we did not directly measure the nature of the INP in our samples. Of course, different biogenic aerosol particles are also

generated differently, but in general, we state the forest with its vegetation as the main INP source, releasing biogenic particles like pollen and fungal spores e.g. through wind dispersion.

SECTION 2 METHODS The methods describing measurements of INPs is quite thorough.
I would like the authors to expand on the uncertainty description for INPs - it is not clear where the systematic error percentages were derived. Were blanks collected?

Thanks for this suggestion. We have added a more detailed description of the determination of the systematic errors to Section 2.2 "INSEKT":

A systematic error due to the preparation process and flow measurements is added. Applying a simple linear error propagation to the formulas given in Vali (1971) and inserting the error-containing parameters like the pipetted suspension volumes and the flow rate systematic errors of 4% for the undiluted suspension, 5% for the first dilution step, 8% for the third dilution step and 11% for the fourth step, are calculated. The systematic error increases with each dilution step because the additional pipetting step adds uncertainty.

In the following sentence, it is mentioned that blank filters have also been collected and the resulting INP temperature spectra have been subtracted from the INP temperature spectra of the "normal" samples.

The inlet for the "additional instrumentation" was heated and "RH remains below 40%"
– how does this compare to the inlet used for the INP filter collections? Without knowing the exact RH at this site, I suspect RH can get quite high and may increase collection efficiencies of large particles.

Thanks for this comment, this is a good point to consider. We do not expect the collection efficiency to be impacted by the relative humidity. However, it is possible that the measured surface area distribution is somehow shifted, which is a common problem when using different inlets. This adds uncertainty to the INAS densitiy parameterization. Currently, we are not able to quantify this uncertainty, but it is important to be aware of the potential bias. It is important to note that the INSEKT is only measuring ice-nucleating particles, which are not soluble.

For the other measurements, there seems to be a few missing details: - What exactly was measured by the L-ToF-AMS? What sizes are detectable by this measurement? - Was the WIBS connected to a similar inlet as the other instruments? Where large super micron particles measurable? Could the authors include a comparison between the scattering particle size distribution measured from the WIBS and the APS data?

We agree with the referee that the paper would benefit from a more detailed explanation on the additional instrumentation used for this study. For both instruments, the WIBS and the L-Tof-AMS, we added a more detailed description, which addresses your questions and also the associated questions of the other referees.

We added a more detailed description on the WIBS instrument to Section 2.3. "Additional Instrumentation at SMEARII":

[revised manuscript text omitted]

We also adjusted the discussion on the comparison of WIBS data to INP in the same Section as follows:

Such a correlation is also supported by the peaks of pollen and PBAP concentrations observed in snow-free periods in spring and in autumn, and by the increases of the organic aerosol mass concentration and fluorescent particle numbers of group ABC observed in spring. According to the study of Savage et al. (2017), we assume particles of fluorescence group ABC to be mainly pollen, particles containing PAH or Ergosterol, or fungal spores.

We also added a more detailed description on the L-ToF-AMS instrument to Section 2.3. "Additional Instrumentation at SMEARII":

The size-resolved chemical composition of ambient aerosol was measured with the L-ToF-AMS. Its application in the same campaign has been described in Paramonov et al. (2020). It builds on the functionality and characteristics of the high-resolution ToF-AMS (DeCarlo et al., 2006). However, due to the longer time-of-flight chamber, the L-ToF-AMS, has a better resolution (8000 M/ΔM) than the standard ToF-AMS (2000 M/ΔM in V-mode). Detailed descriptions of the instrument, measurements and data processing are available in other publications (Canagaratna et al., 2007; DeCarlo et al., 2006). In general, the L-ToF-AMS measures the size-resolved, non-refractory composition of submicron aerosols, including organic, sulfate, nitrate, ammonium and chloride. The aerodynamic lens has a 100% transmission range of 75-650 nm (in vacuum aerodynamic diameter; Liu et al. (2007)) and focuses particles into a narrow beam that impacts the surface of a porous tungsten vaporizer heated to 600$^{\circ}$C, followed by ionization by a 70eV electron source. Ions are detected by a long time-of-flight mass analyzer (Tofwerk AG). The sample flow of 0.09 l min$^{-1}$ is extracted from an extra suction flow (3 l min$^{-1}$) that is used to avoid aerosol losses in the inlet line. A PM2.5 cyclone mounted at the inlet removes large particles to avoid clogging the critical orifice (100µm), and before entering the L-ToF-AMS, the samples are dried by a Nafion dryer to keep the RH below 30%.

The L-ToF-AMS data were analyzed using standard ToF-AMS data analysis toolkits (Squirrel V1.61B and PIKA1.21B) using Igor Pro software (V6.37, WaveMetrics Inc.). To calculate mass concentrations an ionization efficiency (IE) was determined using 300 nm, size-selected, dry ammonium nitrate particles, and a relative ionization efficiency (RIE) for ammonium of 3.7 was determined. The default relative ionization efficiency (RIE) values of 1.1, 1.2, 1.3 and 1.4 for nitrate, sulfate, chloride and organics, respectively, were applied. A composition-dependent collection efficiency (CE) was applied based on the principles proposed by Middlebrook et al. (2012).

We also added the following text to the discussion on Figure 4 (old Figure 3b) to Section 3.2 "Comparison to meteorology ad aerosol properties" to give more information about the measured organics:

The non-refractory organic components measured by the AMS include the commonly observed primary organic aerosol (POA) and oxygenated organic aerosol (OOA).

WIBS vs. APS: We have included a new Figure in the Appendix (Figure A3) showing the size distributions measured with the APS/DMPS (red) and the WIBS (blue) from 11 March 2018 to 13 May 2018. In each panel, one size distribution per month is highlighted to better show the agreement between WIBS and APS/DMPS measurements.
A description of this Figure has been added to section 2.3 "Additional Instrumentation at SMEARII":

Daily size distributions measured by the DMPS and APS combination are compared with the size distributions measured by the WIBS from 11 March 2018 to 13 May 2018 and are shown in Figure A3. Note, that the WIBS only measures particles larger than 0.5 µm. In summer, WIBS tends to measure slightly more particles with diameters larger than about 3 µm compared to the APS. However, the size distributions agree well for the other time periods and the smaller size ranges.

SECTION 3 RESULTS AND DISCUSSION Figure 1 is nearly impossible to read, consider expanding this figure to take up the full page and also increase font sizes for the axes.

We have increased the size of the axis description in Figure 1.

L200-204 – are the increases in INPs, organic aerosol, and fluorescent particles statistically significant? I'm not sure I would consider the organic and fluorescent particle concentrations to demonstrate "clear increases".

We agree that the increase in organic and fluorescent particle concentrations are not as pronounced as the increase in INP concentration. We have therefore reformulated the description:

Both, the organic aerosol mass concentration measured by L-ToF-AMS and the number concentration of PM10 fluorescent particles measured by WIBS, tend to increase show clear increases during the transition period from winter to summer (Fig. 4a and b).

Figure 3 – It looks as though the INP data are smoothed significantly in Figure 3a. This is especially clear when one compared the subfigure (Figure 3b) with the rest of the timeline. what kind of curve is fit to the INP data in Figure 3a? If there is no curve, why is it so smoothed? The daily data or monthly averages as points with standard deviations would be a better, more clear, way of visualizing the data.

INP data are represented as monthly averages in Figure 3a, as are the other data (NPF events, pollen and other PBAP). We have adapted the suggestion to not draw a smoothed line, but to draw points of the averaged INP concentrations with the standard deviation as error bars. Thanks for this suggestion!

L208 – This section is title" Comparison to meteorology and aerosol properties", yet only temperature and snow cover are compared. Are there other ambient variables that are correlated, for example relative humidity & winds? It is possible that relative humidity or winds may impact emissions?

Thanks for this comment. Other referees are also interested in comparisons with other meteorological variables, and therefore we have extended Figure 4 (in the new version, this is Figure 5) by including three additional panels showing the time series of RH (Fig. 5c), wind speed (Fig. 5d) and precipitation (Fig. 5e) and have ajusted the description accordingly. We have included the following section in the manuscript:

Figure 5c shows the comparison of the INP time series with the time series of relative humidity (RH) measured 35 m above ground. Over the entire time period, no clear relationship between RH and INP concentrations is observed. For a shorter time period from June 2018 to September 2018, there seems to be some correlation of the INP concentration with the RH, during which the peaks in INP concentration in June corresponds to a RH peak. In Figure 5d, we compare the time series of INP

concentrations with the time series of wind speed measured 34 m above the ground, which is also above the forest canopy. A relationship between measured INP concentrations and wind speed is not observed. In Figure 5e, the time series of INP concentration is compared to the occurrence of precipitation. Although other studies like Prenni et al. (2013), Huffman et al. (2013) and Iwata et al. (2019) report increasing INP concentrations during and after rain events in forested sites, we do not consistently observe this behavior.  In this respect, increased INP concentrations are observed only during two of the strongest precipitation events in June and September 2018. However, it should be noted that in the cited studies INP concentrations were measured with higher time resolutions from minutes to hours. Huffman et al. (2013) reported increased INP and biological particle concentrations during rain events and up to one day after rain events. With our sampling (and therefore averaging) time of 24 hours or more, rain-induced enhancements of INP concentrations may have been missed. Therefore, this type of sampling strategy may not be appropriate to deterministically link INP concentrations with rain events.

Finally, when considering the full time period of this study, we did not find clear correlations of the INP concentrations with other parameters like the relative humidity, the wind speed or precipitation events. We do not exclude that the INP concentrations measured in the boreal forest are influenced by precipitation events for the whole year. However, our sampling set up is not appropriate to investigate the relationship more precisely.

L258 – While heating does remove a significant portion of the ice nucleation activity, I think it is important for the authors to also acknowledge that the heat-resistant residual is still associated with significant INP numbers. That is, the INP population is not just heat-labile and may not be entirely biological. I think the way the manuscript is currently written indicates that all INPs are biological, but that is not supported by Figure 5, where there are still significant INPs remaining after heating the samples.

Thanks for this comment. As the INP concentrations are significantly reduced after heat-treatment (about two orders of magnitude lower), we concluded that the majority of INPs in our samples are heat-sensitive, what indicates biogenic origin. However, we agree that this does not mean that **all** INPs have to be of biogenic origin. We therefore choose not state that **all** INPs are biogenic, but that the **majority** seems to be of biogenic origin or that boreal forest INP populations **seem to be dominated** by biogenic particles. To address this more clearly in the manuscript, we have added the following text to Section 3.3 "Heat treatment tests":

As a significant number of residual heat-resistant INPs is still remaining after heat treatment, this indicates that not all measured INPs are associated with heat-labile biogenic materials. However, the majority of the INP population seems to be dominated by heat-labile materials, which is shown by the systematic shift in INP-temperature spectra.

L272 – ": : : as they do not include seasonal dependencies." – this is not necessarily the case. Seasonal variability in INP abundances is accounted for by being linked to aerosol amount (in these cases n500 or nFBAP), which have seasonal variability.

That is correct, thank you. We have removed the second part of this sentence:

The predictions of DeMott et al. (2010) and Tobo et al. (2013) overestimate INP concentrations especially in wintertime, .

L267 - I understand the WIBS was used to determine the fluorescent biological aerosol particles (FBAPs); How does this method compare to the UV-APS used in the Tobo et al. (2013) study? Given the number of uncertainties associated with measuring fluorescent measurements, can the authors describe the possible differences between the UV-APS and WIBS and how that would impact the performance of the Tobo et al. (2013) parameterization? Also, how does the Boreal forest in this study differ from that measured in Tobo et al. (2013)?

In general, the UV-APS operates with only one light source (excitation wavelength of 355 nm) and one detection channel (wavelength range of 420-575 nm), which reduces the classification power compared to WIBS. The excitation wavelengths of WIBS are 280nm and 370nm, with detection channels at 310 – 400 nm and 420 – 650 nm and thus the wavelengths of the two instruments also do not agree. For the application of the Tobo et al. (2013) parameterization, we used the WIBS data with excitation wavelength of 370nm and detection channel of 420 – 650nm (FL3), as this comes closest to the set-up of the UV-APS used in Tobo et al. (2013). Besides the elastic light scattering and fluorescence, the UV-APS measures the Aerodynamic particle size (which WIBS does not). Further, UV-APS does not perform any scheduled background "empty chamber" fluorescence check, instead it must be manually checked and calibrated using manual laser power and PMT gain adjustments. Furthermore, the UV-APS is discontinued and TSI no longer supports the instrument.

Concerning the forest around the measurement sites, both are dominated by pine trees and are far from anthropogenic sources. In both studies, it is assumed that the abundant aerosol population is mainly influenced by the forest and emissions from the vegetation. Differences are that the location in Tobo et al. (2013) is in Colorado, America and therefore at lower latitude than the Finnish forest. The dominant pine species in this lower latitudes is ponderosa pine. The Finnish forest is dominated by a different pine species called Scots pine.

Finally, potential differences in the Tobo et al. (2013) and our study, which could impact and explain the performance of the Tobo et al. (2013) parameterization, are the different wavelengths used in the UV-APS and the WIBS, the different latitudes, the different pine species or simply the season in which the measurements have been conducted. The parameterization in Tobo et al. (2013) is based on measurements conducted in the North American monsoon season from July to August 2011.  The HyICE-2018 campaign took place from March to May 2018. In this time period, we do not observe a relation of INP concentrations to FL3 fluorescent particle concentrations. It cannot be excluded that this relation becomes stronger in the summer months, as it is observed in Tobo et al. (2013).

Figure 6 – I think it would be helpful to mention that only data from the HyICE-2018 period were possible to use in the Tobo et al. (2013) (2) parameterization and therefore that panel has fewer points.

Thanks for this suggestion. We have added the following text to Section 3.4 Parameterizations:

Note for the application of the Tobo et al., (2013) parameterization using FBAP concentrations, only data from the HyICE-2018 time period could be used, as the fluorescence measurements from the WIBS are only available in this period. Therefore, the number of data points in Fig. 7c is lower than in the other panels.

L295 – Should clearly state that this parameterization is based on ground-level ambient temperature, not "ambient temperature". If T is described as ambient temperature, a model will implement this as

the temperature predicted at any level, which I do not think is the intention. Additionally, while I understand the impressive correlation between
INP and ambient temperature is inviting for a parameterization, I caution the authors in publishing this as an INP parameterization given that it is highly specific to this location and has not been tested for other years, or is not really supported by a physical mechanism.
For example, aerosol-based INP parameterizations have a physical mechanism
– an aerosol particle that is seasonally variable with an assumed ice nucleation density.
The second parameterization presented in this paper has more physical meaning, linking INP abundance to the physical process of snow melting and exposed surface emissions. Without a clear physical basis for a "ground-level temperature-based" parameterization, I would recommend removing this.

The "ground-level temperature-based" parameterization is an empirical description of the INP concentration at ground level. It is not a theory, which tries to describe the physical mechanisms and processes behind the observed INP. Thus, this parameterization is not an explanation of the complex processes that result in observed INP occurrences. This formulation can still be useful as a technical basis for atmospheric models, which could help to improve the representation of atmospheric INP concentration by using the ground-level ambient temperature. The models will not manage to describe the real physical processes behind aerosol particle behaviors as this is very complex and any model cannot resolve the involved time scales. We therefore suggest this parameterization to technically describe INP concentrations in models and to test it and develop it in further studies.

We added "ground-level" to all passages where we discuss the ambient air temperature to make clear, that we do not use ambient air temperature in other higher altitudes. Thanks for this comment!

L311 – What is T in this equation?

Thanks for this question. T is the activation temperature of the ice-nucleating particles. As this explanation is missing in the manuscript, we have added the following text right after Eq. (2):

[…], where T is the activation temperature of INPs.

**References:**

[revised manuscript text omitted]

---

## Referee Report (RR1)

The revised manuscript by Schneider et al. has significantly improved from the original version. I only have a few minor comments below that the authors should address prior to acceptance.

I appreciate that the authors took care in adding more sufficient background. Perhaps a bit long in some places. I suggest shortening the discussion on Hartmann et al. to a 2 – 3 sentences so it is similar in length to the others.

The paragraph starting on page 1 is quite lengthy. The authors should consider splitting up into multiple, more focused paragraphs, e.g., one on previous work, one on boreal forest background, and one on biogenic INPs. That leads nicely into the second (last) paragraph that provides a synopsis of the current work.

Page 5, line 208: Why was only the one inlet heated? Would that not introduce variability that would affect the sizing results?

Again, the paragraph starting at the end of page 5 is quite long. Consider breaking up into size distributions (DMPS+APS), fluorescence measurements (WIBS), and chemical measurements (L-ToF-AMS).

Page 6, line 239: Can eliminate the sentence starting with "Note" as that is redundant.

Section 3.2: Another long paragraph. Consider splitting up to make more concise paragraphs.

Page 9, lines 293 – 303: This information belongs in the methods section.

Page 10, lines 341 – 343: Wouldn't this be expected given the wind measurements were above the canopy and the INP measurements were below?

Page 16, line 452: What a remarkable reproduction of the data! Very great result.

---

## Author Response (AR2)

**Response to the Editor Decision:**

Dear authors,

Thank you very much for your revised version and your reply to the reviewer's comments. There are a few things to be corrected before your manuscript can be accepted for ACP:

1.) Reviewer 3 (and also reviewer 1) commented on the parameterization of INP number concentration via temperature. Although this is an interesting finding, I agree with the reviewers that the physical meaning is missing. The mathematical relationship is not really useful to be directly implemented into models. I asked an additional reviewer (with modelling expertise of INP) for an additional opinion and the reviewer agrees with reviewer 3 that the related text should be revised. It has to be clear that the found relationship is useful to constrain models (please also revise the abstract & conclusions accordingly) but it is not a direct parameterization. See detailed comments for reviewer 3 and 4.

2.) Reviewer 1 asked a question about the vegetation indices (LAI, NDVI). I found this to be a useful additional information to the manuscript. You could add this comparison at least to the supplement and add a sentence to the result section.

3.) Reviewer 2 had some more minor revisions.

Looking forward to your final revisions!

Many thanks and kind regards

Paul Zieger.

Dear Paul Zieger,

Thank you for your feedback! We really appreciate your effort to include an additional referee in this discussion. We found his/her comments very useful and helpful.

Regarding point 1): We agree, that we need to point out more clearly, that this is not a direct parameterization of INP concentrations in the atmosphere, like other common parameterizations that are functions of aerosol number or surface area concentration, but more a new way to formulate the annual variability of the INP concentration in a larger source region like the boreal

forest in our case. There, the annual variation of the ambient INP concentration may be controlled or at least largely influenced by temperature dependent source processes. To account for this, we followed the suggestions of Referee #4 and modified the manuscript accordingly. Please see the response to Referee #4 for details.

Regarding point 2): We looked again into the data of NDVI and PRI and we still think that there is no benefit from this comparison to the story of the manuscript. We observe increases of NDVI and PRI during the transition period from winter to summer, but this does not happen in the same time period when we observe increasing INP concentrations. For now, we do not have a profound basis to relate our INP observations to the NDVI and PRI variability throughout the year. A thorough analysis and interpretation would raise and involve more questions, which are beyond the scope of this manuscript and need further detailed data analysis, which will be the subject of a follow-up study.

Regarding point 3): We followed the suggestions of Referee #2 and modified the manuscript accordingly. For details, please see the response to Referee #2.

Thanks again and best regards,

Julia Schneider, on behalf of all authors

**Response to the referee report by Referee #2:**

We thank referee #2 for his or her helpful feedback. Please find below our responses and suggestions for the manuscript revision, with the referee comments in black, our answers in red and suggested changes or additions to the manuscript in blue.

The revised manuscript by Schneider et al. has significantly improved from the original version. I only have a few minor comments below that the authors should address prior to acceptance.

I appreciate that the authors took care in adding more sufficient background. Perhaps a bit long in some places. I suggest shortening the discussion on Hartmann et al. to a 2 – 3 sentences so it is similar in length to the others.

Thanks for this suggestion. We adjusted the discussion on the study of Hartman et al. (2019) as follows:

Hartmann et al. (2019) report INP concentrations from the past 500 years derived from ice core samples collected at two Arctic sites.  They do suggest indications that biological INPs contribute to Arctic INP populations throughout the past centuries,  and assume that it is likely that the strength of local biological particle sources is enhanced during a particular time of the year influencing the INP variability. However, due to the time resolution and dating uncertainty a seasonal relation could not be explicitly shown.

The paragraph starting on page 1 is quite lengthy. The authors should consider splitting up into multiple, more focused paragraphs, e.g., one on previous work, one on boreal forest background, and one on biogenic INPs. That leads nicely into the second (last) paragraph that provides a synopsis of the current work.

We follow the referee's suggestion and split the respective paragraph into shorter parts: general introduction and previous work (first), boreal forest environment (second), biogenic INPs (third), information about this study (fourth).

Page 5, line 208: Why was only the one inlet heated? Would that not introduce variability that would affect the sizing results?

The heated inlet is a total aerosol inlet for the size distribution measurements, which are permanently installed at SMEARII. The heating of the inlet to about 40°C is important to also sample fog or cloud droplets when they are present at the site. Such droplets can be larger than 10 μm in diameter, and then do not pass the PM10 inlets, which we use for the INP filter sampling. In such a situation, we may underestimate the INP number concentration compared to a total aerosol inlet. At dry conditions without large droplets present, which was the case most of the time during this study in Hyytiälä, the difference between the inlets is negligible. Therefore, the dry aerosol measured at the heated total aerosol inlet is a good reference for our INP measurements at the PM10 inlet. In general, it is better to measure both INPs and aerosol parameters at the same inlet, but this was not

possible during this study in Hyytiälä because of space and inlet flow limitations. The effect of different inlets, with different size cutoffs, and parallel INP sampling or measurements with size-selected aerosols are a topic of ongoing and upcoming studies.

Again, the paragraph starting at the end of page 5 is quite long. Consider breaking up into size distributions (DMPS+APS), fluorescence measurements (WIBS), and chemical measurements (LToF-AMS).

We split this section into three paragraphs, as suggested.

Page 6, line 239: Can eliminate the sentence starting with "Note" as that is redundant.

We removed this sentence from the manuscript.

Section 3.2: Another long paragraph. Consider splitting up to make more concise paragraphs.

We split Section 3.2 in four shorter paragraphs. In principle, three Figures (Fig. 3,4 and 5) are discussed in this Section. Therefore, we split the text in one paragraph for each Figure description (three paragraphs in total) and a fourth paragraph where the findings of the three Figures are contextualized.

Page 9, lines 293 – 303: This information belongs in the methods section.

We have shifted the information to the Methods Section "2.3 Additional instrumentation at SMEARII". We have also added a short description of the other instruments that measure meteorological variables at SMEARII and are used in this study.

We have added the following text:

Various meteorological parameters are continuously monitored at SMEARII. For this study, we used five basic variables including ambient air temperature, relative humidity, wind speed, snow depth and precipitation. The ambient air temperature was measured 4.2 m above ground with a Pt100 sensor inside a ventilated custom-made radiation shield. This 4.2 m temperature measurement is the closest to ground-level at SMEARII and thus we utilize this as the ground-level ambient air temperature in the following. The relative humidity was measured in 35 m height by a Rotronic MP102H RH sensor. For wind speed measurements, we used a Thies 2D Ultrasonic anemometer at 34 m above the ground by. The snow depth was measured by a Jenoptik SHM30 snow depth sensor, which is based on an opto-electronic laser distance sensor, in open field about 500 m southeast of the aerosol collection area of SMEARII. The precipitation, the liquid water equivalent, was measured by a Vaisala FD12P Weather Sensor in 18 m height.

Page 10, lines 341 – 343: Wouldn't this be expected given the wind measurements were above the canopy and the INP measurements were below?

During the intensive measurements period during the HyICE-2018 campaign from March to May 2018, additional INP measurements were conducted on a 35 m high tower to directly compare INP concentrations above and below the forest canopy. These measurements were not included in this study, as they were only available for a short period of the year and did not encompass a seasonal cycle. However, we observed no systematic difference between the INP concentrations below and above the canopy. Therefore, it was not implicitly expected that there is no relationship between INP concentrations below the canopy and the wind speed above the canopy observed.

More detailed analyses and additional comparisons of INP concentrations below and above the canopy will be included in future work, which will focus on the intensive measurement period of the HyICE-2018 campaign.

Page 16, line 452: What a remarkable reproduction of the data! Very great result.

Thank you!

**References:**

Hartmann, M., Blunier, T., Brügger, S. O., Schmale, J., Schwikowski, M., Vogel, A., Wex, H. and Stratmann, F.: Variation of Ice Nucleating Particles in the European Arctic Over the Last Centuries, Geophys. Res. Lett., 46(7), 4007–4016, doi:10.1029/2019GL082311, 2019.

**Response to the referee report by Referee #3:**

We thank referee #3 for his or her feedback. Please find our responses below and suggestions for the manuscript revision, with the referee comments in black, our answers in red, and suggested changes or additions to the manuscript in blue.

The authors have identified a strong correlation between the ground temperature and the INP number concentrations, which they interpret as a link to seasonality. This correlation is then used to "develop a parameterization". In my original comment, I emphasize how, while the parameterization "works", it doesn't have a physical meaning and also wouldn't be something realistic to implement in a model. I think I would be ok if the authors want to present the analysis as a fit to data in supplemental material, but I don't think it belongs in the main text nor do I think it's a valid parameterization.

The authors respond "This formulation can still be useful as a technical basis for atmospheric models, which could help to improve the representation of atmospheric INP concentration by using the ground-level ambient temperature." - This is not a reasonable application, as this would require a model that includes tracers to determine the aerosol origin when calculating INP numbers at cloud level (this is not a common framework and is also very computationally expensive). I think the authors may not be familiar with how models estimated ice nucleation and INPs, as the models do not track INPs from various sources, but only represent the emissions of aerosols and the physical processes that act on aerosols (e.g., scavenging); Ice nucleation is then calculated based on a parameterization applied to a simulated aerosol parameter (e.g., abundance of biological particles, Hoose et al. 2010).

That is, the INP number concentration that is measured at the surface will not be the same as at cloud-level. Meaning, if using this ground-based-temperature "parameterization", one needs to have additional information on things like the number & type of particles to know how to scale the INP number concentrations at higher altitudes. That is, if you have an airmass that originates from this sampling location with e.g., ground temperature of 10 deg C and a corresponding number concentration of INP active at 257K (nINP) of 0.3 per liter, once that air mass reaches cloud level, the nINP will be altered due to 1) dilution, 2) scavenging, 3) mixing with other air masses from different location containing different aerosol.

Finally, the authors write "The models will not manage to describe the real physical processes behind aerosol particle behaviors as this is very complex and any model cannot resolve the involved time scales.", which is not correct. Aerosol models have become increasingly sophisticated and are able to represent number, size, and mass of aerosols from many sources. Model timescales are as capability of resolving aerosol processes as they are cloud processes. For the biological particles that appear dominate in the INP population in this study, for example, there are a many studies to illustrate growing modeling capabilities, for which I provide a couple:

Model-estimated bioaerosol evaluated against single particle mass spectrometer measurements

Zawadowicz, M. A., Froyd, K. D., Perring, A. E., Murphy, D. M., Spracklen, D. V., Heald, C. L., et al. (2019). Model-measurement consistency and limits of bioaerosol abundance over the continental United States. Atmos. Chem. Phys., 13.

Classical nucleation theory based approach for estimating IN, including biological particles:

Hoose, C., Kristjánsson, J. E., Chen, J.-P., & Hazra, A. (2010). A Classical-Theory-Based Parameterization of Heterogeneous Ice Nucleation by Mineral Dust, Soot, and Biological Particles in a Global Climate Model. Journal of the Atmospheric Sciences, 67(8), 2483–2503. https://doi.org/10.1175/2010JAS3425.1

All in all, I stand by my comment that the ground temperature based "parameterization" should not be included in the main text or presented as a parameterization.

We note the referee's concerns and critical comments about the temperature-based parameterization. After discussion with the co-authors and consideration of the comments from other referees, we think there is justification for the presentation of the parameterization within the manuscript, given a few clarifications and modifications. We agree that the background for and applicability of this parameterization was not well explained and have therefore added explanatory statements. For example, as suggested by Referee#4, we explain that this is a mechanistic parameterization, which describes the near-surface INP concentrations as a function of the temperature, assuming that this concentration is dominated by temperature-dependent INP sources. In addition, we clarify within the text that this mechanistic parameterization only describes INP concentrations near the surface, and that it is only applicable in the boreal forest area. With this, we hope to clarify how this formulation can be used and applied in models.

We have changed the following passages, as shown below. For more details of the suggested changes to the manuscript, please see also the response to Referee #4.

Abstract: As current parameterizations do not reproduce this variability, we suggest a new mechanistic  description for boreal forest environments, which considers the seasonal variation of INP concentrations. For this, we use the  ambient air temperature measured close to the ground at 4.2 m height as a proxy for the season, which appears to affect the source strength of biogenic emissions and thus the INP abundance over the boreal forest .

Methods: The ambient air temperature was measured 4.2 m above ground with a Pt100 sensor inside a ventilated custom-made radiation shield. This 4.2m temperature measurement is the closest to ground-level at SMEARII and thus we utilize this as the ground-level ambient air temperature in the following.

Description of Equ. (1) (Sec. 3.4): To account for seasonal dependencies in this formulation, the linear relation between the  ambient air temperature $T_{amb}$ in K measured close to the ground at 4.2 m height (called ground-level ambient air temperature) and the natural logarithm of the time series of INP concentrations [...].

[…] with a1 = 0.074 ± 0.006, a2 = -18 ± 2, b1 = -0.504 ± 0.005, b2 = 127 ± 1 and with the activation temperature T and ground-level ambient air temperature $T_{amb}$ in K (measured at 4.2 m height).

Discussion on parameterization (Sec. 3.4): ~~With this new approach, we do not directly describe the INP concentration in the atmosphere in a specific environment, as it was common in previous studies (DeMott et al., 2010; Tobo et al., 2013). Rather we have found a way to describe the boreal forest as an important INP emitting source. We suggest this formulation for application in atmospheric models to describe the source concentration of INPs abundant at ground level, which can then be further transported to cloud-relevant altitudes within the models. Here, it needs to be considered that~~ This new parameterization approach describes the annual variation of the near-surface INP concentration in the boreal forest, which provides a temperature dependent source of these INPs. We did not directly detect or quantify the INP source, but found a strong correlation of the measured INP concentration with the ground-level ambient air temperature. This leads to the assumption that the near-surface INP concentration in the boreal forest may be dominated by local or regional sources, and that this parameterization may be used in models to formulate the source strength or concentration of INPs in boreal forests near the surface. It should be noted that this is a mechanistic approach, which cannot necessarily be applied to regions other than boreal areas or to higher altitudes, where the INP spectrum may be influenced by other sources. It is further important to note that INPs might undergo changes in their size distribution and chemical composition, when they are transported from their sources to higher altitudes, which could affect their ice nucleation ability.

Conclusions: Thus, we suggest two new approaches for formulating and quantifying the annual  variability of INP concentrations over boreal forest areas. The first is a mechanistic approach, which considers the boreal forest as  a temperature dependent INP emission source with a pronounced seasonal cycle.

**Response to the referee report by Referee #4:**

We thank referee #4 for his or her thoughtful comments and feedback. Please find below our responses and suggestions for the manuscript revision, with the referee comments in black, our answers in red, and suggested changes or additions to the manuscript in blue.

I have been asked to comment on a specific point raised by Reviewer#3, but also mentioned by Reviewer#1 : the question of the feasibility of the temperature dependent parameterization.

This parameterization shows very impressive results when compared to observations. This suggests that the temperature close to the surface might represent the variability in measured INP concentrations better than any other meteorological parameter (such as humidity in the surface layer) or any index representative of the vegetation. I agree with Reviewer#3 that a physical explanation of this relationship is missing, but I wouldn't remove this temperature-dependent parameterization. The possible contribution of PBAPs to INP population is probably somewhat embedded in this relation. I still believe it is valuable for models because INP concentrations are generally very badly represented, or even not taken into account (many atmospheric models just use a temperature and supersaturation dependent parameterization). It could help to improve the representation of atmospheric INP concentrations and their seasonal variability. Hence, I think it should be kept in the paper. But, to avoid confusion for modelers who could be interested in this parameterization, it should be clearly stated that :

1/ what is called ambient temperature in the pre-exponential factor is the air temperature close to the surface. In models, it is generally represented by the 2m temperature;

2/ this is not a physical parameterization, but a mechanistic parameterization to represent sources near the ground level that may impact INP populations;

3/ this parameterization is only relevant over boreal forest areas, not elsewhere.

To 1): We did not use the air temperature directly measured at the surface, but in 4.2 m above the surface, which is the temperature measurement closest to the surface and continuously available at SMEARII. Therefore, we called this ground-level temperature here.

We made this clear point clearer in the manuscript by adding the information about real temperature measurement height of 4.2 m into the Methods section as well as in the description of equation (1), which gives the parameterization.

Methods: The ambient air temperature was measured 4.2 m above ground with a Pt100 sensor inside a ventilated custom-made radiation shield. This 4.2m temperature measurement is the closest to ground-level at SMEARII and thus we utilize this as the ground-level ambient air temperature in the following.

Description of Equ. (1) (Sec. 3.4): To account for seasonal dependencies in this formulation, the linear relation between the  ambient air temperature $T_{amb}$ in K measured close to the ground

at 4.2 m height (called ground-level ambient air temperature) and the natural logarithm of the time series of INP concentrations […].

[…] with a1 = 0.074 ± 0.006, a2 = -18 ± 2, b1 = -0.504 ± 0.005, b2 = 127 ± 1 and with the activation temperature T and ground-level ambient air temperature $T_{amb}$ in K (measured at 4.2 m height).

To 1), 2) and 3): We followed your suggestions and adjusted the corresponding passages in the text to clarify that we used the ambient air temperature measured close to the ground, that the parameterization is only valid in boreal environments and that the parameterization is only mechanistic, as follows:

Abstract: As current parameterizations do not reproduce this variability, we suggest a new mechanistic  description for boreal forest environments, which considers the seasonal variation of INP concentrations. For this, we use the  ambient air temperature measured close to the ground at 4.2 m height as a proxy for the season, which appears to affect the source strength of biogenic emissions and thus the INP abundance over the boreal forest .

Discussion on parameterization (Sec. 3.4): ~~With this new approach, we do not directly describe the INP concentration in the atmosphere in a specific environment, as it was common in previous studies (DeMott et al., 2010; Tobo et al., 2013). Rather we have found a way to describe the boreal forest as an important INP emitting source. We suggest this formulation for application in atmospheric models to describe the source concentration of INPs abundant at ground level, which can then be further transported to cloud-relevant altitudes within the models. Here, it needs to be considered that~~ This new parameterization approach describes the annual variation of the near-surface INP concentration in the boreal forest, which provides a temperature dependent source of these INPs. We did not directly detect or quantify the INP source, but found a strong correlation of the measured INP concentration with the ground-level ambient air temperature. This leads to the assumption that the near-surface INP concentration in the boreal forest may be dominated by local or regional sources, and that this parameterization may be used in models to formulate the source strength or concentration of INPs in boreal forests near the surface. It should be noted that this is a mechanistic approach, which cannot necessarily be applied to regions other than boreal areas or to higher altitudes, where the INP spectrum may be influenced by other sources. It is further important to note that INPs might undergo changes in their size distribution and chemical composition, when they are transported from their sources to higher altitudes, which could affect their ice nucleation ability.

Conclusions: Thus, we suggest two new approaches for formulating and quantifying the annual  variability of INP concentrations over boreal forest areas. The first is a mechanistic approach, which considers the boreal forest as  a temperature dependent INP emission source with a pronounced seasonal cycle.

---

## Author Response (AR3)

**Response to the Editor Decision:**

Dear authors,

Thanks for the revisions. Your publication is almost ready for publication after the following minor issues have been corrected:

- The mathematical description of Eq. 1 and 2 is not fully correct. Please add correct units to fit coefficients and equation (i.e. that the correct concentration unit follows from the right side of the equation). It would also be helpful for the reader to see goodness of fit parameters (e.g. R-squared) for the your new mechanistic parameterization (e.g. in Fig. 8 and/or within the text).

- Maybe also make clear that these are empirical equations that do not follow a physical law (e.g. you could re-name 3.4 to "Empirical parameterizations").

- Units in Fig A3 are missing (y-axis).

Thanks and kind regards

Paul.

Dear Editor,

Thank you for your feedback! We revised the manuscript according to your suggestions. Here are the details:

Regarding point 1): For Equ. (1) and (2), we added the correct units to the equations as well as to the fit coefficients, where needed. Further, the goodness of fit for the new mechanistic parameterization with $R^2$ = 0.82, and the goodness of fit for the new INAS density parameterizations with $R^2$ = 0.78, 0.78 and 0.85 were added to the text (see lines 464 – 466 and lines 488 - 489).

Regarding point 2): We renamed the section 3.4. to 'Empirical parameterizations', as suggested.

Regarding point 3): We added the unit 'in $cm^{-3}$' to the y-axis label in Fig. A3.

Thanks again and best regards,

Julia Schneider, on behalf of all authors